# VT-Bench: A Unified Benchmark for Visual-Tabular Multi-Modal Learning

**Zi-Yi Jia** [1 2]   **Zi-Jian Cheng** [1 2]   **Xin-Yue Zhang** [1 2]   **Kun-Yang Yu** [1 3]   **Zhi Zhou** [1]   **Yu-Feng Li** [1 3]   **Lan-Zhe Guo** [1 2 ✉]

## Abstract

Multi-model learning has attracted great attention in visual-text tasks. However, visual-tabular data, which plays a pivotal role in high-stakes domains like healthcare and industry, remains underexplored. In this paper, we introduce *VT-Bench*, the first unified benchmark for standardizing vision-tabular discriminative prediction and generative reasoning tasks. VT-Bench aggregates 14 datasets across 9 domains (medical-centric, while covering pets, media, and transportation) with over 756K samples. We evaluate 23 representative models, including unimodal experts, specialized visual-tabular models, general-purpose vision-language models (VLMs), and tool-augmented methods, highlighting substantial challenges of visual-tabular learning. We believe VT-Bench will stimulate the community to build more powerful multi-modal vision-tabular foundation models.

Benchmark: https://github.com/LAMDA
-NeSy/VT-Bench

## 1. Introduction

Multi-modal foundation models have achieved remarkable success in ***visual-text*** tasks. Conversely, another important multi-model setting, ***visual-tabular*** learning, has received far less attention. Vision-tabular data is prevalent in high-stakes domains such as healthcare (Cui et al., 2023) and industrial analytics (Huang et al., 2022), where the two modalities provide complementary and often non-substitutable information. For example, in medical diagnosis, images (e.g., radiology) convey lesion morphology and spatial patterns, while structured tables (e.g., laboratory tests) provide precise physiological data and numerical measurements. Successfully integrating these heterogeneous signals is critical for achieving both high performance and clinical reliability (Acosta et al., 2022; Huang et al., 2020a).

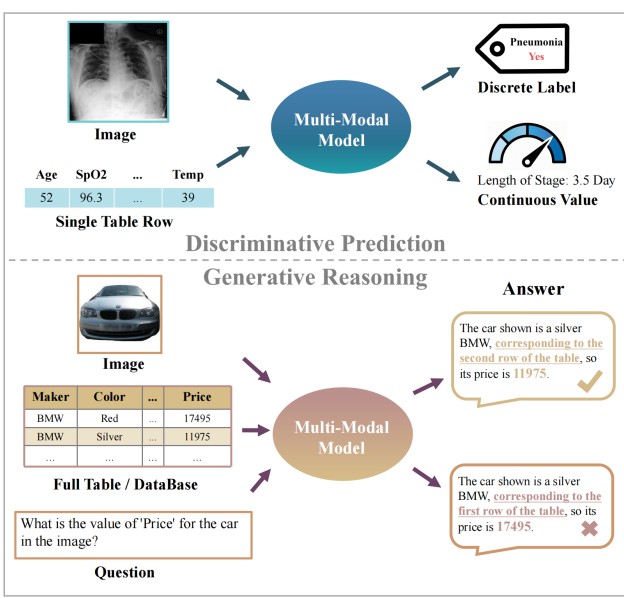

*Figure 1.* Two paradigms in vision–tabular multi-modal learning and the cross-modal grounding challenge in generative reasoning.

As shown in Figure 1, visual-tabular multi-modal learning includes two paradigms:

- **Discriminative Prediction**, which learns a function that maps an input sample, including an image and a tabular row, to a discrete label for classification or a continuous value for regression.
- **Generative Reasoning**, where the model performs multi-step reasoning over the input image and tabular data to answer a question with rationales.

These two learning paradigms emphasize different capabilities: discriminative prediction focuses on robust multi-modal feature aggregation, while generative reasoning requires precise evidence localization and complex logical composition. Consequently, a comprehensive benchmark should cover both paradigms to fully characterize the challenges of vision-tabular multi-modal learning.

Existing vision–tabular benchmarks remain limited in both scope and evaluation focus. In discriminative prediction,

---

[1]National Key Laboratory for Novel Software Technology, Nanjing University, China [2]School of Intelligence Science and Technology, Nanjing University, China [3]School of Artificial Intelligence, Nanjing University, China. Correspondence to: Lan-Zhe Guo <guolz@nju.edu.cn>.

prior benchmarks, such as RadFusion (Zhou et al., 2021), are confined to narrowly defined subfields, limiting their generality. In generative reasoning, recent benchmarks assess vision–tabular capabilities by rendering tables as images (Kim et al., 2024; Singh et al., 2025), which shifts reasoning almost entirely to the visual modality and fails to evaluate multi-modal evidence integration. Meanwhile, multi-modal table reasoning benchmarks model vision–tabular interactions by placing images into table cells (Mathur et al., 2024; Titiya et al., 2025). Still, their protocols largely overlook structure-aware understanding and constrained numerical reasoning over two-dimensional tables in multi-modal setting, which are capabilities commonly viewed as essential in TableQA. Consequently, the field lacks a general-purpose benchmark that supports systematic evaluation of vision–tabular learning across both discriminative and generative settings.

To bridge the aforementioned gaps, this paper presents the first systematic study and empirical analysis of vision–tabular multi-modal learning, and introduces VT-Bench, the first general-purpose vision–tabular benchmark that jointly covers both discriminative prediction and generative reasoning tasks. Our contributions are summarized as follows:

- **Unified Vision-Tabular Benchmark.** VT-Bench contains 14 datasets across 9 domains with over 756K samples, providing a rigorous evaluation of 23 representative models. Furthermore, we construct 4 novel datasets to bridge critical evaluation gaps in clinical prediction, tabular structure perception, and constrained numerical reasoning.

- **Holistic Evaluation Framework with Novel Diagnostics.** Beyond reporting six standard metrics for discriminative prediction, we introduce two modality-specific diagnostic metrics, Modality Contribution Ratio (MCR) and Modality Informativeness Ratio (MIR), which enable a fine-grained dissection of fusion dynamics by quantifying modality reliance and disentangling visual and tabular contributions.

- **Empirical Findings and Methodological Implications.** Through extensive experimentation, we derive several critical insights regarding current model capabilities:
  - **Fusion Bottlenecks:** Existing fusion strategies often fail to reliably integrate heterogeneous visual and tabular representations, frequently resulting in **negative transfer**, where multi-modal integration degrades performance compared to unimodal baselines.
  - **Reasoning Deficits in VLMs:** VLMs exhibit significant limitations in visual grounding, constrained statistical computation, and program generation. Notably, open-source models suffer further from a lack of robust **tabular structure perception**.

Building on these findings, we summarize the core challenges of vision-tabular multi-modal learning, analyze their root causes, and provide practical recommendations for future model design and research.

## 2. Related Works

**Vision-Tabular Multi-Modal Methods.** Vision–tabular multi-modal models have predominantly been developed for discriminative prediction tasks. Early work follows a late-fusion paradigm, using dual encoders that process images and tabular features independently before combining them through simple operator-level fusion, such as Concat (Spasov et al., 2019), Max (Vale-Silva & Rohr, 2021), or MUL (Duanmu et al., 2020). Subsequent methods move toward earlier and more interactive fusion. For instance, DAFT (Pölsterl et al., 2021) introduces conditional affine transformations to modulate visual features with tabular signals, while CHARMS (Jiang et al., 2024) uses optimal-transport-based alignment to better match cross-modal representations. More recent models, such as MMCL (Hager et al., 2023) and TIP (Du et al., 2024), incorporate contrastive learning and pretraining objectives to further enhance interaction and consistency between visual and tabular representations. Despite these developments, comparisons across studies remain challenging due to varying datasets and evaluation protocols. Conversely, in generative reasoning tasks, although some fields have developed specific workflows based on VLMs(Bae et al., 2023), these methods have limited applicability, and there is still a lack of models that support general vision-tabular reasoning.

**Benchmark for Vision-Tabular Multi-Modal Learning.** Prior work on vision-tabular multi-modal learning has produced a growing set of datasets and evaluation resources. For discriminative prediction, systematic benchmarking remains scarce: benchmarks such as RadFusion (Zhou et al., 2021) typically adopt protocols tailored to narrow subfields (e.g., specific clinical scenarios) and emphasize in-domain aggregate performance, which limits generality and detailed diagnosis. For generative reasoning, several benchmarks combine visual inputs and tabular information. A common protocol renders tables as images (e.g., TableVQA-Bench (Kim et al., 2024) and MTabVQA (Singh et al., 2025)), mainly measuring visual document understanding and thereby weakening the evaluation of alignment and evidence integration when vision and structured tables are treated as separate sources (Jiang et al., 2026). Another line of work models vision-tabular interactions through multi-modal table reasoning by embedding images as table-cell contents, as in MMTabQA (Mathur et al., 2024) and MMT-Bench (Titiya et al., 2025). However, current evaluations do not systematically probe table-structure understanding or constrained numerical reasoning in vision-tabular settings, despite these being central capabilities in the TableQA literature (Jiang et al., 2025; Ji et al., 2026). Consequently, VT-Bench is the first unified, multi-domain benchmark to jointly cover discriminative prediction and generative reasoning tasks, with protocols that explicitly evaluate the key capabilities required for reliable vision-tabular learning.

## 3. Task and Notation

### 3.1. Task Setting

This paper studies vision-tabular multi-modal learning problem, where each sample comprises at least one image $I$ and associated structured tabular data $S$. Depending on the content of $S$ and the desired output format, we categorize the task space into two main paradigms: *discriminative prediction* and *generative reasoning*.

### 3.2. Discriminative Prediction

In discriminative prediction, the structured input $S$ is represented as a single-row tabular feature vector $x$, aligned with an image $I$. The objective is to predict a target variable $y$, which can be a discrete label $y \in \{1, \ldots, K\}$ for classification or a continuous value $y \in \mathbb{R}^p$ for regression. The dataset is defined as $\mathcal{D}_{\text{pred}} = \{(I_i, x_i, y_i)\}_{i=1}^N$. The model learns a function $f_\theta$ that maps the joint image and tabular inputs to the target:

$$\hat{y} = f_\theta(I, x).$$

### 3.3. Generative Reasoning

Generative reasoning tasks require the model to answer a natural language question $q$ based on visual input and structured information. The output is a textual answer $a$. This task commonly appears in two forms.

**(1) Table-form reasoning.** The structured input $S$ is a full table $T$. The dataset is $\mathcal{D}_{\text{tbl}} = \{(I_i, T_i, q_i, a_i)\}_{i=1}^N$, and the model learns $\hat{a} = f_\theta(I, T, q)$. Some reasoning datasets additionally provide optional textual evidence $c$ (e.g., retrieved passages), in which case the mapping becomes $\hat{a} = f_\theta(I, T, q, c)$.

**(2) Database-form reasoning.** The structured input $S$ is a relational database $B$. In this setting, the dataset is denoted as $\mathcal{D}_{\text{db}} = \{(I_i, B_i, q_i, a_i)\}_{i=1}^N$, and the model outputs $\hat{a} = f_\theta(I, B, q)$. Compared to table-form reasoning, database-form typically requires the model to generate an executable SQL query over $B$ to retrieve relevant records, and then reason over the resulting table to produce the answer.

## 4. VT-Bench

VT-Bench is a unified benchmark for vision-tabular multi-modal learning, covering 14 datasets across discriminative prediction and generative reasoning, and evaluating 23 representative models. In addition, VT-Bench introduces a comprehensive metrics suite for analyzing vision-tabular fusion in discriminative prediction tasks, including six overall metrics and two modality-level diagnostics, MCR and MIR. It also provides a unified and reproducible Python API to support systematic evaluation and future comparisons.

### 4.1. Discriminative Prediction

#### 4.1.1. DATASETS

To systematically evaluate the modality fusion and generalization performance of vision–tabular multi-modal models, we select 8 open-source and reliable datasets covering binary classification, multi-class classification, and regression across multiple high-value domains. Dataset attributes are summarized in Table 1, with full dataset descriptions and preprocessing details in Section E.2.1.

In addition, our preliminary survey indicates that existing medical vision–tabular discriminative prediction datasets are limited in both scale and task diversity, which constrains evaluation in realistic clinical settings. To address this limitation, we construct three large-scale medical vision–tabular datasets based on MIMIC-IV v2.2 (Johnson et al., 2023) and MIMIC-CXR v2.0.0 (Johnson et al., 2019). These datasets cover clinical discriminative prediction tasks from diagnosis to prognosis, including one classification task (Pneumonia) and two regression tasks (Length of Stay and Respiratory Rate). Detailed dataset construction procedures and task definitions are described in Section E.2.2.

#### 4.1.2. MODELS

To systematically characterize the strengths and limitations of distinct modeling paradigms, we evaluate three categories of methods: unimodal baselines, vision-tabular multi-modal models, and VLMs. Model details and hyperparameter settings are provided in Section F.

**Unimodal Models.** Unimodal results serve as references for analyzing multi-modal gains and as empirical proxies for modality informativeness. For images, we adopt two representative baselines, ResNet-50 (He et al., 2016) and ViT-B/16 (Dosovitskiy et al., 2021). For tabular data, we adopt LightGBM (Ke et al., 2017) as a tree-based baseline, and include TabTransformer (Huang et al., 2020b) and TabPFN-v2 (Hollmann et al., 2025) to represent deep tabular models and in-context learning approaches.

**Vision-Tabular Multi-modal Models.** We evaluate two fusion paradigms: late fusion and early-interaction fusion, distinguished by whether cross-modal signals influence representation learning during encoding. Late fusion include Concat (Spasov et al., 2019), MAX (Vale-Silva & Rohr, 2021), and MUL (Duanmu et al., 2020). Early-interaction models include DAFT (Pölsterl et al., 2021) and CHARMS (Jiang et al., 2024), as well as contrastive pretraining methods such as MMCL (Hager et al., 2023) and TIP (Du et al., 2024).

**General-Purpose VLMs.** We include Qwen3-VL-8B-Instruct (Bai et al., 2025) and Table-LLaVA-v1.5-7B (Zheng et al., 2024) as representative VLM baselines and fine-tune them using supervised fine-tuning (SFT) on each task. Ta-

*Table 1.* Summary of discriminative prediction datasets in VT-Bench. We list dataset metadata and MIR, a unimodal-performance proxy for relative modality informativeness; see Section 4.1.3.

| Dataset | Domain | Target | #Samples | #Cat. | #Num. | #Labels | MIR |
|---|---|---|---|---|---|---|---|
| **Classification** | | | | | | | |
| Breast Cancer | Oncology | Breast cancer diagnosis | 3,287 | 5 | 1 | 4 | 0.9513 |
| Skin Cancer | Dermatology | Skin disease prediction | 2,298 | 19 | 3 | 6 | 1.4510 |
| Infarction | Cardiology | Myocardial infarction prediction | 44,245 | 26 | 49 | 2 | 0.9521 |
| Pneumonia | Pulmonology | Pneumonia detection | 74,064 | 2 | 11 | 2 | 0.9734 |
| Adoption | Pets | Pet adoption category | 14,652 | 1 | 24 | 5 | 1.1149 |
| CelebA | Vision | Attractive attribute prediction | 202,599 | 39 | 0 | 2 | 0.9744 |
| DVM-Car | Transportation | Vehicle type classification | 176,414 | 4 | 13 | 286 | 1.1182 |
| **Regression** | | | | | | | |
| Los | Pulmonology | Length-of-stay prediction | 64,048 | 4 | 14 | ＼ | 1.0415 |
| Respiratory Rate | Pulmonology | Respiratory rate prediction | 73,243 | 2 | 13 | ＼ | 0.5321 |
| Pawpularity | Pets | Pet popularity score prediction | 7,930 | 8 | 0 | ＼ | 1.0180 |
| Anime | Media | Anime rating prediction | 12,513 | 5 | 6 | ＼ | 1.7435 |

bles are serialized in markdown, and prompt templates are provided in Section E.

### 4.1.3. METRICS

For discriminative prediction tasks, we use standard holistic metrics: Accuracy, AUC, and Macro-F1 for classification, and RMSE, MAE, and $R^2$ for regression. Beyond holistic performance, we introduce two modality diagnostic metrics, MCR and MIR, to analyze fusion and decision behaviors at a finer granularity. MCR measures the model's dependence on each modality, while MIR estimates the relative informativeness of each modality at the dataset level and provides a data-prior reference for interpreting MCR.

**Modality Contribution Ratio.** MCR quantifies modality dependency by replacing one modality with an uninformative reference during inference and measuring the resulting performance drop. We adopt the mean embedding computed over the training set as this non-informative reference (e.g., replacing the vision embedding $h_I$ with the mean vision embedding $\bar{h}_I$ for vision ablation, and the same for tabular ablation). To unify classification and regression tasks, we define an error metric $\mathcal{E}$: we set $\mathcal{E} = 1 - \text{ACC}$ for classification and $\mathcal{E} = \text{RMSE}$ for regression. Let $\mathcal{E}^{\text{full}}$ denote the error under full multi-modal input, and $\mathcal{E}^{\text{abl}(I)}$, $\mathcal{E}^{\text{abl}(T)}$ be the errors after image and table ablation. The variations are: $\Delta_I = \mathcal{E}^{\text{abl}(I)} - \mathcal{E}^{\text{full}}$, $\Delta_T = \mathcal{E}^{\text{abl}(T)} - \mathcal{E}^{\text{full}}$. Here, $\Delta_m > 0$ ($m \in \{I, T\}$) indicates that removing modality $m$ increases the error, implying a positive contribution, whereas $\Delta_m < 0$ suggests that the modality introduces interference and causes negative transfer. To enable cross-task comparison, we normalize the variation between modalities and define the Modality Contribution Ratio:

$$\text{MCR}_I = \frac{\Delta_I}{|\Delta_I| + |\Delta_T|}, \qquad \text{MCR}_T = \frac{\Delta_T}{|\Delta_I| + |\Delta_T|}.$$

The sign of $\text{MCR}_m$ indicates the contribution direction, while its magnitude reflects the relative impact, providing a decision-level measure of modality dependency. We assess the robustness of MCR to the choice of non-informative reference and training-data size in Section D.

**Modality Informativeness Ratio.** MIR measures the relative predictive informativeness of the two modalities at the dataset level, using the error of the best unimodal baseline as a proxy. Let $\mathcal{E}^*_{\text{img}}$ and $\mathcal{E}^*_{\text{tab}}$ denote the optimal unimodal errors for vision and tabular models, respectively. MIR is defined as

$$\text{MIR} = \frac{\mathcal{E}^*_{\text{img}}}{\mathcal{E}^*_{\text{tab}}}.$$

MIR > 1 indicates that the tabular modality is more informative, while MIR < 1 suggests vision dominance; MIR ≈ 1 implies comparable informativeness. We treat MIR as a dataset-level prior that, combined with model-centric MCR, supports analysis of fusion and decision behavior. The robustness of MIR to proxy choices is examined in Section D.

### 4.2. Generative Reasoning Tasks

#### 4.2.1. DATASETS

**EHRXQA.** EHRXQA (Bae et al., 2023) is one of the few medical QA benchmarks for "the Chest X-ray + structured EHR" setting. Evidence is implicitly stored in relational databases rather than provided as explicit vision–tabular pairs, requiring structured retrieval followed by cross-modal medical reasoning. This design reflects real clinical workflows and evaluates both retrieval planning and reasoning capabilities of VLMs.

In VT-Bench, we evaluate EHRXQA using a two-stage pipeline and report end-to-end accuracy. In the first stage,

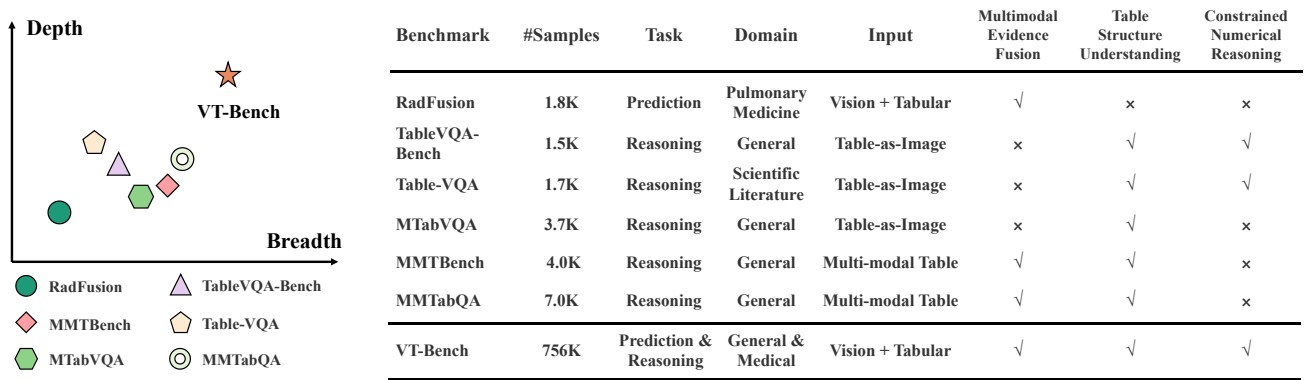

| Benchmark | #Samples | Task | Domain | Input | Multimodal Evidence Fusion | Table Structure Understanding | Constrained Numerical Reasoning |
|---|---|---|---|---|---|---|---|
| RadFusion | 1.8K | Prediction | Pulmonary Medicine | Vision + Tabular | √ | × | × |
| TableVQA-Bench | 1.5K | Reasoning | General | Table-as-Image | × | √ | √ |
| Table-VQA | 1.7K | Reasoning | Scientific Literature | Table-as-Image | × | √ | √ |
| MTabVQA | 3.7K | Reasoning | General | Table-as-Image | × | √ | × |
| MMTBench | 4.0K | Reasoning | General | Multi-modal Table | √ | √ | × |
| MMTabQA | 7.0K | Reasoning | General | Multi-modal Table | √ | √ | × |
| VT-Bench | 756K | Prediction & Reasoning | General & Medical | Vision + Tabular | √ | √ | √ |

*Figure 2.* The comparison between VT-Bench and existing vision–tabular benchmarks. VT-Bench achieves stronger breadth by covering both discriminative prediction and generative reasoning paradigms while spanning diverse domains, and greater depth by evaluating key capabilities for vision–tabular learning.

the model produces a structured execution plan, including whether retrieval is needed and the corresponding SQL query, which is executed to obtain patient sub-tables and chest X-rays. In the second stage, the execution plan, retrieved evidence, and questions are provided to the model to generate the final answer. To separate retrieval and reasoning errors, we additionally conduct a decoupled evaluation. Stage 1 evaluates the correctness of retrieval planning and SQL generation, while Stage 2 evaluates answer generation given retrieved evidence. Detailed dataset descriptions and implementation details for the decoupled evaluation are provided in Section E.3.1.

**Multi-ModalQA.** Multi-ModalQA (MMQA) (Talmor et al., 2021) is a multi-modal QA benchmark integrating textual, tabular, and visual evidence to evaluate multi-source aggregation and multi-hop compositional reasoning. We report end-to-end accuracy under the original setting as a holistic reference. Dataset details and evaluation settings are provided in Section E.3.2.

**DVM-Car QA.** Existing vision–tabular reasoning benchmarks lack a dedicated setting for natural images paired with tables, leaving key table-QA capabilities, such as table-structure understanding and logically constrained numerical reasoning, underexplored in multi-modal settings.

To fill this gap, we construct DVM-Car QA, a fine-grained vision–tabular reasoning benchmark based on the DVM-Car dataset. Each instance is a triplet $(I, T, q)$, where a front-view car image $I$ is paired with a table $T$ containing $n \in \{10, 20, 50\}$ candidate vehicles (15 fixed attributes), and exactly one row uniquely corresponds to the car in $I$ via a *Vision Alignment Key* (e.g., color). This design makes both modalities necessary: models must first extract visual cues from the image to localize the target row in the table, and then derive the final answer solely from the table. The benchmark consists of four progressively challenging tasks:

(1) *Row Localization*, identifying the target row via visual cues; (2) *Cell Retrieval*, extracting a specified attribute from the target row; (3) *Constrained Counting*, filtering and counting entries under target-relative conditions; and (4) *Conditional Mean*, computing the mean of a numerical attribute over filtered samples. We also add an *Identification* diagnostic to disentangle failures in visual cue extraction versus table-structure understanding by verifying whether the model's explanation names the unique *Vision Alignment Key*. Together, these tasks evaluate cross-modal alignment, tabular structure understanding, and constrained numerical reasoning. By varying alignment cues and table sizes, DVM-Car QA further probes model robustness under different perceptual and computational loads. Detailed construction procedures and task definitions are provided in Section E.3.3.

We further analyze the evidence modalities required for sample-level decisions across VT-Bench in Section E.1. The results show that VT-Bench covers diverse evidence requirements and that most predictions or question answers require integrating visual and tabular evidence.

### 4.2.2. MODELS

As no widely adopted standard architecture has yet emerged for vision–tabular generative reasoning, we evaluate the zero-shot performance of general-purpose VLMs and tool-augmented methods in the "vision + table/database" setting. We consider six open-source VLMs: InternVL3-8B (Zhu et al., 2025), Qwen3-VL-8B-Instruct (Yang et al., 2025), Qwen3-VL-8B-Thinking (Yang et al., 2025), GLM-4.1V-9B-Thinking (Hong et al., 2025), Llama-3.2-11B-Vision-Instruct (Patterson et al., 2022), and Pixtral-12B (Agrawal et al., 2024a), together with two proprietary models, GPT-4.1(OpenAI, 2025) and Gemini-3-Flash-Preview(Google DeepMind, 2025). We also include two representative tool-augmented reasoning methods, Thyme (Zhang et al., 2025) and StructGPT (Jiang et al., 2023). For StructGPT, to enable

*Table 2.* Results on VT-Bench across two task families, evaluating three representative model families.

*(a)* Discriminative prediction result. We report Accuracy for classification and RMSE for regression. The best overall result is **bold**, and the best among vision-tabular multi-modal models is underlined. Rank denotes the average per-dataset rank across all datasets. Relative performance gap (Δ) is defined as the average performance gap to the best unimodal model. **Green** indicates improvement, whereas **red** indicates degradation.

| Method | Rank | Classification (↑) | | | | | | | | Regression (↓) | | | | |
| | | Breast Cancer | Skin Cancer | Infarction | Pneumonia | Adoption | DVM-Car | CelebA | Δ | Los | Respiratory Rate | Pawpularity | Anime | Δ |
|---|---|---|---|---|---|---|---|---|---|---|---|---|---|---|
| **Unimodal Baseline** | | | | | | | | | | | | | | |
| ResNet-50 | 9.5 | 0.224 | 0.370 | 0.670 | 0.719 | 0.302 | 0.856 | 0.822 | ↘ | 18.808 | 9.102 | 21.188 | 0.840 | ↘ |
| ViT-16 | 7.1 | **0.574** | 0.578 | **0.964** | 0.677 | 0.371 | 0.879 | 0.784 | ↘ | 14.644 | **4.192** | 21.189 | 0.869 | ↘ |
| LightGBM | 6.1 | 0.546 | **0.839** | 0.709 | 0.670 | 0.410 | 0.983 | 0.792 | ↘ | **14.060** | 12.141 | 21.238 | 0.497 | ↘ |
| TabPFN v2 | **5.3** | 0.493 | 0.835 | 0.726 | 0.697 | **0.414** | 0.913 | 0.801 | ↘ | 18.111 | 7.878 | 20.813 | 0.482 | ↘ |
| Tabtransformer | 5.7 | 0.507 | 0.741 | 0.917 | 0.699 | 0.392 | 0.944 | 0.789 | ↘ | 18.486 | 9.075 | 20.815 | 0.528 | ↘ |
| **Vision–Tabular Multi-modal Models with Early-Interaction Fusion** | | | | | | | | | | | | | | |
| CHARMS | 8.1 | 0.040 | 0.452 | 0.853 | 0.722 | 0.286 | 0.940 | 0.813 | -0.126 | 18.367 | 9.111 | 19.924 | 0.912 | 1.855 |
| MMCL | 8.9 | 0.313 | 0.370 | 0.647 | 0.724 | 0.307 | 0.912 | 0.826 | -0.127 | 20.114 | 9.131 | 20.854 | 0.866 | 0.757 |
| TIP | 8.5 | 0.461 | 0.715 | 0.728 | 0.682 | 0.309 | 0.984 | 0.800 | -0.045 | 21.603 | 9.660 | 21.218 | 0.541 | 1.435 |
| DAFT | 7.7 | 0.412 | 0.674 | 0.699 | 0.723 | 0.292 | 0.975 | 0.822 | -0.056 | 18.878 | 9.042 | 21.324 | 0.598 | 0.235 |
| **Vision–Tabular Multi-modal Models with Late Fusion** | | | | | | | | | | | | | | |
| MAX | 6.5 | 0.503 | 0.752 | 0.699 | 0.721 | 0.312 | 0.955 | 0.824 | -0.032 | 18.219 | 9.151 | 21.273 | 0.532 | 0.215 |
| Concat | 5.4 | 0.503 | 0.748 | 0.721 | 0.722 | 0.334 | 0.953 | 0.828 | -0.026 | 18.062 | 9.218 | 20.922 | 0.535 | -0.557 |
| Mul | 9.5 | 0.503 | 0.729 | 0.693 | 0.720 | 0.280 | 0.961 | 0.821 | -0.040 | 18.209 | 21655801.1 | 161.841 | 6.475 | 5413983.6 |
| **VLMs** | | | | | | | | | | | | | | |
| Table-LLaVA-v1.5-7B | 9.1 | 0.040 | 0.374 | 0.703 | 0.699 | 0.362 | 0.4434 | 0.826 | -0.220 | 19.415 | 13.946 | 23.050 | 0.463 | 0.312 |
| Qwen3-VL-8B-Instruct | 7.5 | 0.303 | 0.709 | 0.7425 | 0.715 | 0.389 | 0.968 | 0.823 | -0.049 | 19.824 | 9.204 | 21.627 | **0.436** | 1.925 |

*(b)* Generative reasoning result. The best result is shown in **bold**, and the best open-source model is underlined.

| TASK# | Open-Source General Models | | | | | | Proprietary Models | | Tool-Augmented Methods | |
| | InternVL3-8B | Qwen3-VL-8B Instruct | Qwen3-VL-8B Thinking | GLM-4.1V-9B Thinking | Llama-3.2-11B Vision-Instruct | Pixtral-12B | GPT-4.1 | Gemini-3-Flash-Pre. | StructGPT | Thyme |
|---|---|---|---|---|---|---|---|---|---|---|
| **DVM-Car QA** | | | | | | | | | | |
| Identification | 0.7033 | 0.9056 | 0.9567 | **0.9578** | 0.5589 | 0.8856 | 0.8178 | 0.8956 | 0.7211 | 0.7511 |
| Row Localization | 0.4567 | 0.6756 | 0.6133 | 0.7511 | 0.4189 | 0.5289 | 0.7300 | **0.9533** | 0.8578 | 0.4361 |
| Attribute Retrieval | 0.3389 | 0.4978 | 0.5733 | 0.6900 | 0.2489 | 0.6200 | 0.8844 | **0.9300** | 0.4956 | 0.6722 |
| Constrained Counting | 0.2533 | 0.4233 | 0.0189 | 0.4200 | 0.1922 | 0.1600 | 0.4167 | **0.5111** | 0.3000 | 0.2244 |
| Conditional Mean | 0.0433 | 0.0578 | 0.0230 | 0.2056 | 0.0433 | 0.0856 | 0.2111 | **0.4967** | 0.0067 | 0.1389 |
| Average Accuracy | 0.2731 | 0.4136 | 0.3071 | 0.5167 | 0.2258 | 0.3486 | 0.5606 | **0.7228** | 0.4150 | 0.3679 |
| **MMQA** | | | | | | | | | | |
| TableQ | 0.7073 | 0.6694 | 0.5550 | 0.7260 | 0.3740 | 0.7019 | 0.7556 | **0.7922** | 0.7178 | 0.6450 |
| TextQ | 0.7101 | 0.7060 | 0.6684 | 0.6946 | 0.4355 | 0.7420 | 0.7300 | **0.7470** | 0.6876 | 0.6200 |
| ImageQ | 0.2783 | 0.4783 | 0.3678 | 0.4488 | 0.2391 | 0.4217 | 0.5730 | **0.6447** | 0.5493 | 0.4605 |
| ImageListQ | 0.0922 | 0.2411 | 0.0724 | 0.2478 | 0.0780 | 0.1418 | 0.3137 | **0.3510** | 0.2319 | 0.1739 |
| Multi-Hop TASK | 0.3886 | 0.4171 | 0.2036 | 0.3997 | 0.1961 | 0.4161 | 0.5948 | **0.7234** | 0.3462 | 0.4274 |
| Average Accuracy | 0.4724 | 0.5300 | 0.3924 | 0.5621 | 0.2824 | 0.5339 | 0.6360 | **0.7087** | 0.5411 | 0.5092 |
| **EHRXQA** | | | | | | | | | | |
| FULL | 0.2219 | 0.1920 | 0.0000 | 0.0728 | 0.0830 | 0.1666 | 0.2643 | **0.2751** | 0.1416 | 0.0440 |
| STAGE 1 | 0.6017 | 0.5206 | 0.0000 | 0.2317 | 0.2900 | 0.6181 | **0.7937** | 0.6297 | 0.4626 | 0.2410 |
| STAGE 2 | 0.4052 | 0.4449 | 0.2716 | 0.3854 | 0.3842 | 0.3796 | 0.4628 | **0.5157** | 0.3080 | 0.1820 |

multimodal input, we replaced its default backbone with Qwen3-VL-8B-Instruct.

All experiments use official public checkpoints and recommended inference settings, without fine-tuning. For database-form reasoning, we convert the relevant database schema and required metadata into text and include them in the prompt. We serialize tables in Markdown format and adopt consistent decoding strategies across all reasoning datasets. We do not render tables as images because many tables are large (e.g., 50 × 15) and would require very high resolution to keep text legible, substantially increasing the number of visual tokens. More importantly, table images make results sensitive to model-specific visual processing and visual text recognition, which would confound the evaluation of table reasoning in vision-tabular multi-modal setting. Model details and configurations are provided in Section F.4.

### 4.2.3. METRICS

We evaluate models by exact-match accuracy, as our generative tasks use relatively canonical short answers, such as row indices. Given predictions $\hat{a}_i$, the accuracy is defined as:

$$\text{Acc} = \frac{1}{N} \sum_{i=1}^{N} \mathbf{1}\big[\text{canon}(\hat{a}_i) = \text{canon}(a_i)\big].$$

Here, $\text{canon}(\cdot)$ applies standard normalization (e.g., ignoring punctuation, extra spaces, and invisible characters).

### 4.3. Comparisons with Existing Benchmarks

To further distinguish the difference between VT-Bench and other existing ones, we elaborate the benchmark details in Figure 2. From the breadth perspective, prior benchmarks often focus on a single paradigm in vision-tabular learning,

*Table 3.* MCR of multi-modal models on VT-Bench discriminative prediction datasets. Values in red indicate modalities exhibiting negative transfer. For each dataset, the value in parentheses reports its MIR. Definitions and computation of MCR and MIR are provided in Section 4.1.3. The VLMs reported in this table correspond to Qwen3-VL-8B-Instruct and Table-LLaVA-v1.5-7B.

| | Breast Cancer (0.9513) | | Skin Cancer (1.4510) | | Infarction (0.9521) | | Pneumonia (0.9734) | | Los (1.0415) | | Resp. Rate (0.5321) | | Adoption (1.1149) | | DVM-Car (1.1182) | | CelebA (0.9744) | | Pawpularity (1.0180) | | Anime (1.7435) | |
|---|---|---|---|---|---|---|---|---|---|---|---|---|---|---|---|---|---|---|---|---|---|---|
| | Img | Tab | Img | Tab | Img | Tab | Img | Tab | Img | Tab | Img | Tab | Img | Tab | Img | Tab | Img | Tab | Img | Tab | Img | Tab |
| TIP | 16.7 | 83.3 | 0.0 | 100.0 | 29.0 | 71.0 | -49.4 | -50.6 | 12.8 | 87.2 | -87.7 | -12.3 | 0.0 | 100.0 | 12.7 | 87.3 | 79.6 | 20.4 | -3.4 | 96.6 | 13.6 | 86.4 |
| Max | 0.5 | 99.5 | 0.0 | 100.0 | 0.0 | -100.0 | 18.5 | 81.5 | 29.1 | 70.9 | -6.8 | -93.2 | 0.0 | 100.0 | 56.4 | 43.6 | 75.7 | 24.3 | 10.2 | -89.8 | 4.0 | 96.0 |
| Concat | 0.9 | 99.1 | 0.0 | 100.0 | -23.3 | -76.7 | 0.0 | 100.0 | 40.3 | 59.7 | -19.6 | -80.4 | -80.1 | 19.9 | 60.6 | 39.4 | 86.7 | 13.3 | 51.6 | 48.4 | -0.1 | 99.9 |
| DAFT | -57.1 | 42.9 | 0.0 | 100.0 | -60.8 | 39.2 | 23.6 | 76.4 | 28.7 | 71.3 | -11.1 | 88.9 | -15.8 | 84.2 | 10.5 | 89.5 | 82.1 | 17.9 | 98.9 | 1.1 | 17.3 | 82.7 |
| Qwen3-VL | 28.9 | 71.1 | 44.1 | 55.9 | 64.5 | 35.5 | 11.9 | 88.1 | 65.6 | 34.4 | 43.0 | 57.0 | 18.2 | 81.8 | 7.2 | 92.8 | 0.2 | 97.8 | -41.1 | 58.9 | 93.1 | 6.9 |
| Table-LLaVA | 0.0 | -100.0 | -46.1 | -53.9 | 36.3 | 63.7 | 11.9 | 88.1 | 53.4 | 46.6 | 4.2 | 95.8 | 99.2 | 0.8 | 56.2 | 43.8 | 16.5 | 83.5 | 71.0 | 29.0 | 98.4 | 1.6 |

either discriminative prediction or generative reasoning, and typically concentrate on general everyday knowledge or a narrow domain. In contrast, VT-Bench spans 9 scenarios, covering representative medical specialties, as well as everyday domains. Regarding evaluation depth, existing benchmarks commonly suffer from two limitations. First, many present tables in a purely visual form (i.e., table-as-image), where vision and tabular modality largely convey the same evidence, making it difficult to assess perception and reasoning over multi-source evidence. Second, benchmarks based on multi-modal tables may provide mixed-modal inputs, yet often lack systematic evaluation of table structure understanding and constrained numerical reasoning. By contrast, VT-Bench is designed to comprehensively assess the key capabilities required for vision–tabular tasks.

### 4.4. How to use VT-Bench

We release a unified Python API for VT-Bench to enable standardized and reproducible benchmarking across discriminative prediction and generative reasoning tasks. The API supports unified task definition, dataset selection, model setup, evaluation settings, and optional modality diagnostics, allowing consistent comparison under a single interface while remaining extensible to custom models and datasets. Detailed usage examples and configuration descriptions are provided in Section A.1.

## 5. Results and Analysis

We conducted experiments on VT-Bench to characterize the core challenges of vision-tabular multi-modal learning and to assess model adaptability. As summarized in Table 2, current multi-modal approaches, including both specialized vision-tabular models and general-purpose VLMs, face significant challenges in this setting, with recurring limitations observed across different evaluation scenarios. We further provide model ranking analyses across task domains for discriminative prediction and across capability dimensions for generative reasoning in Section A.3.

In discriminative prediction tasks, multi-modal models do not consistently outperform the strongest unimodal base-

lines across various datasets and task types. In generative reasoning tasks, even state-of-the-art proprietary models show limited performance: accuracy reaches only 72% on DVM-Car QA, 70% on MMQA, and 27% on EHRXQA. To understand these limitations, we conduct a detailed analysis, summarize three key findings, and offer recommendations for future research. Building on these insights, we further summarize the unique challenges, limitations, and suggestions for vision–tabular multi-modal learning in Section C.

> **Finding 1: Prevalence of Negative Transfer.** Both vision-tabular models and VLMs frequently exhibit negative transfer, resulting in performance degradation due to ineffective multimodal fusion.

To obtain a finer-grained view of decision behavior on discriminative prediction tasks, we compute MCR for all models using both modalities during inference (excluding MUL due to regression instability) and estimate dataset-level modality priors via MIR. As shown in Table 3, negative transfer is prevalent across models and datasets: removing one modality often improves performance. Because this phenomenon typically occurs only for a subset of models within the same dataset, it is unlikely to be explained by dataset-level noise. For Respiratory Rate, which shows negative transfer in five models (the highest among our datasets), we conducted additional analysis and observed that removing both modalities produces a substantially higher RMSE (19.240 averaged across models) than using both modalities or either modality alone. This indicates that each modality carries useful information; however, when an additional modality is introduced, the model fails to leverage it as useful evidence and is instead distracted by it.

Notably, the phenomenon of negative transfer extends beyond scenarios where removing a weaker modality is beneficial; in certain settings, removing either modality also improves performance. This contrasts sharply with vision–text discriminative tasks, where modalities typically share high semantic overlap within continuous embedding spaces. Vision–tabular tasks, conversely, involve highly heterogeneous modalities that encode complementary but structurally distinct information. Consequently, existing fusion paradigms

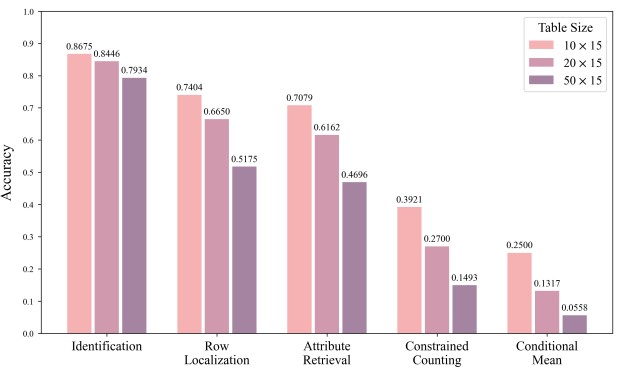

*Figure 3.* Model-averaged accuracy on DVM-Car QA across Identification and four task types under varying sub-table sizes.

struggle to bridge this gap, failing to learn discriminative representations within a shared space and instead causing destructive interference. **Future research should therefore prioritize learning discriminative and separable representations to accommodate such heterogeneity, thereby ensuring more stable and robust fusion.**

Additional analyses of fusion architecture, training strategy, and backbone selection for vision–tabular models are provided in the Section B.

> **Finding 2: Bottlenecks in Vision Modality Perception.**
> Accurate perception and grounding of relevant visual information within long vision–tabular input remains a critical bottleneck.

Vision–tabular generative reasoning can be decomposed into two stages: (1) perception, where the model identifies and extracts the evidence from each modality; and (2) reasoning, where it composes the extracted evidence to derive the final answer. Our experiments indicate that current models are challenged in both stages under the vision–tabular setting.

Table 2 provides evidence that the primary bottleneck lies in perception: when long-context inputs mix text, tables, and images, models often fail to locate and extract the relevant visual evidence. On MMQA, the ImageList and ImageListQ subtasks, which depend primarily on visual information, consistently yield the lowest accuracy. In DVM-Car QA, as shown in Figure 3, increasing the table size from $n$=10 to 50 causes a marked performance drop on Identification tasks. This pattern indicates that longer contexts impair attention to critical visual cues, yielding less stable predictions. Such failures are particularly common in vision–tabular reasoning because tables linearized into text are often highly verbose.

What's more, open-source models show systematic deficits in perceiving and grounding table structure. As shown in Table 2, open-source models exhibit a clear performance decline across perception-related tasks in DVM-

Car QA: Acc(Identification) > Acc(Row Localization) > Acc(Attribute Retrieval). This pattern indicates that while models can often recognize the relevant visual cue, their ability degrades when translating it into precise row positions and extracting cell-level information, highlighting persistent weaknesses in perceiving and grounding table structure.

> **Finding 3: Deficits in Numerical and Symbolic Reasoning.** Constrained numerical reasoning and code generation pose persistent challenges for current models over structured visual-tabular data.

In DVM-Car QA, as shown in Table 2, all models exhibit a clear accuracy drop on *Constrained Counting* and *Conditional Mean*, indicating difficulty in extracting multiple numerical values from large tables and performing constrained arithmetic and statistical operations. Together with the perception and grounding issues in Finding 2, these results suggest that neither prompt-only reasoning over table renderings nor straightforward tool augmentation is sufficient. **Future work should investigate multi-agent approaches that integrate tool-based evidence perception and executable table reasoning to improve the reliability of the vision-tabular reasoning process.**

Experiments on EHRXQA highlight several challenges for deploying current general-purpose VLMs in realistic clinical QA pipelines. In Stage 1, open-source models often fail to translate clinical questions into executable SQL, with the best achieving about 60% success. One plausible contributor is that clinical queries frequently involve multi-table joins with higher-order nesting, together with fine-grained temporal and topological constraints over events and clinical entities. In Stage 2, even when correct evidence is provided, accuracy remains limited, suggesting that clinical semantic reasoning is still challenging under this setup. **A potential direction is neuro-symbolic clinical QA, coupling neural generation with explicit symbolic constraints and execution-based verification to improve robustness across stages.**

## 6. Limitations

Our conclusions are restricted to the evaluated datasets, tasks, and models, and primarily apply to settings with similar modality distributions and cross-modal dependencies. VT-Bench also does not exhaustively cover all tabular representation formats, such as CSV, HTML, SQL, Markdown, and table images, under separate format-wise splits. A broader evaluation of table serialization formats remains valuable future work. In addition, we do not provide a theoretical analysis linking MCR/MIR to model performance; developing a unified framework that connects modality priors, fusion strategies, and generalization across datasets remains an important direction for future work.

## 7. Conclusion

We introduce VT-Bench, a comprehensive benchmark for evaluating vision-tabular multi-modal learning under a unified and standardized protocol, covering both discriminative prediction and generative reasoning tasks across high-impact application domains. To support fine-grained analysis, we propose a metric suite that includes six standard evaluation metrics and two modality-specific diagnostic metrics. Using VT-Bench, we conduct a systematic empirical study, distill three key findings, and identify core challenges together with promising directions for future research. To enhance accessibility and ensure reproducibility, we provide intuitive and user-friendly Python APIs for dataset access, standardized evaluation, and integration into experimental workflows, together with a public leaderboard for transparent comparison and community engagement. We hope VT-Bench will enable reliable comparison, comprehensive diagnosis, and continued progress in vision-tabular multi-modal learning.

## Benchmark Availability Statement

The benchmark code for this paper is publicly available at https://github.com/LAMDA-NeSy/VT-Bench. The project page of VT-Bench, which provides the public leaderboard, additional benchmark information, and relevant supplementary resources, is available at VT-Bench Homepage.

## Acknowledgements

This work was supported by the Key Program of Jiangsu Science Foundation (BK20243012), the National Science Foundation of China (62306133), and the "111 Center" (No. B26023). This research has been conducted using the UK Biobank Resource under Application Number 1142379. We sincerely thank the reviewers for their constructive comments and insightful suggestions.

## Impact Statement

This paper introduces VT-Bench, a unified benchmark aimed at advancing systematic research in vision-tabular multi-modal learning. The empirical observations and diagnostic analyses presented in this work may have broad and multifaceted societal implications, particularly for the evaluation, comparison, and development of reliable multi-modal systems in high-impact application domains. However, given the specific scope and primary objectives of this benchmark study, a comprehensive discussion of downstream ethical considerations, deployment risks, and long-term societal consequences is beyond our present focus and is left to future work. These discussions are expected to center on best practices, responsible evaluation protocols, and practical guidelines that effectively support real-world applications.

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

# VT-Bench: A Unified Benchmark for Visual-Tabular Multi-Modal Learning

## Supplementary Material

## Table of Contents in Appendix

# A. VT-Bench Community Contribution

## A.1. VT-Bench API

To support reproducible evaluation and systematic comparison of vision–tabular methods, we provide a unified Python API for VT-Bench. The API offers a consistent interface for both discriminative prediction tasks and generative reasoning tasks, while allowing straightforward extension to new datasets and models. For discriminative prediction, the API further supports optional modality-level diagnostic metrics to enable fine-grained analysis. The complete implementation and usage details are available at `https://github.com/LAMDA-NeSy/VT-Bench`.

The API is centered around five core arguments:

The `task` argument specifies the evaluation task type, taking values from `prediction`, `reasoning`. The former corresponds to the classification and regression tasks described in Section 4.1, while the latter covers the multi-modal reasoning tasks introduced in Section 4.2.

The `dataset` argument defines the evaluation dataset under the selected task type. All datasets listed in Section E are supported by default, and additional datasets can be incorporated by following the benchmark's standardized data format.

The `model` argument determines the model to be evaluated. Users may select from the built-in baseline models or register custom models through a unified interface. Instructions for adding new datasets and models are provided in the repository documentation under "How to Add New Datasets & Models".

The `setting` argument controls the evaluation configuration. It is mainly used in generative reasoning tasks to select different evaluation stages or analysis modes. For example, in EHRXQA, we provide the options `stage1`, `stage2`, and `full`. For discriminative prediction tasks, this argument can be ignored.

The `diagnostics` argument indicates whether additional modality diagnostic metrics are computed for discriminative prediction tasks. Available options include `none/mcr/mir/full`. This argument has no effect on generative reasoning.

Example commands are shown below:

```
Example Command

 python run.py ---task prediction ---dataset pneumonia ---model TIP ---setting none
 ---diagnostics full
 python run.py ---task reasoning ---dataset ehrxqa ---model Qwen/Qwen3-VL-8B-Instruct
 ---setting ehrxqa_full ---diagnostics none
```

## A.2. Public Leaderboard

To facilitate transparent comparison and long-term community engagement, we plan to maintain a public leaderboard for VT-Bench. The leaderboard is intended to report standardized evaluation results across supported vision–tabular tasks under unified evaluation protocols. As the benchmark evolves, evaluation results for newly introduced models and datasets will be incorporated using the official VT-Bench evaluation pipeline to ensure consistent data splits and metrics. We encourage researchers to contribute results for their own models or datasets. In future updates, we plan to extend the leaderboard to support both public and private usage modes. The project page of VT-Bench, which contains the leaderboard and additional benchmark information, is available at VT-Bench Homepage.

## A.3. Model Ranking Analysis

To provide a more direct and systematic comparison across different models, we further summarize model rankings on both discriminative prediction and generative reasoning tasks. For discriminative prediction, we compute the relative rank of each model within each individual domain and visualize the resulting rankings as a heatmap in Figure 4. For generative reasoning, we aggregate model rankings across seven distinct capability dimensions and present a comprehensive radar-style summary in Figure 5. In both visualizations, a lower numerical rank indicates stronger comparative performance.

In discriminative prediction, strong unimodal models remain highly competitive across several domains, suggesting that multimodal fusion does not uniformly dominate when the predictive signal is sufficiently captured by structured or single-modality features. In contrast, for generative reasoning tasks, Gemini-3-Flash-Preview shows the clearest overall

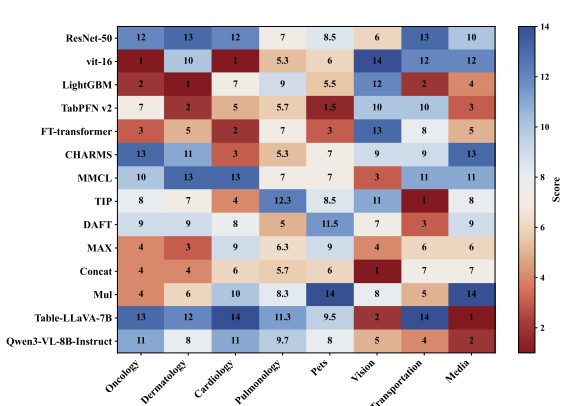

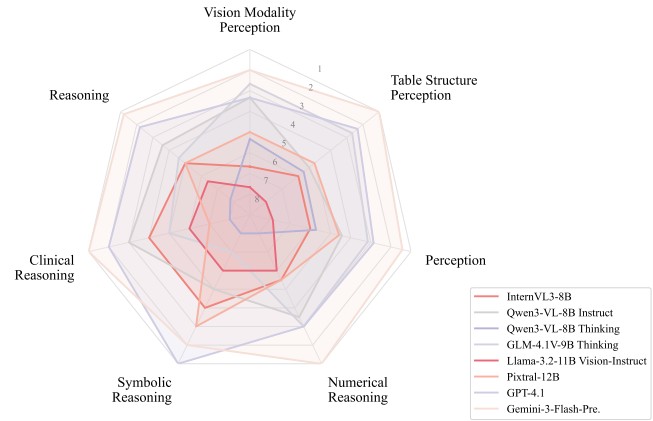

*Figure 4.* Model Ranking across Eight Domains in Discriminative Prediction. Each cell reports the model's rank within a domain, where a lower rank indicates better performance.

*Figure 5.* Radar-style ranking summary of VLM performance across seven capability dimensions for generative reasoning tasks. Lower values indicate stronger relative performance.

advantage across capability dimensions, indicating stronger zero-shot reasoning ability under our evaluation protocol.

## B. Additional Analyses for Vision–Tabular Models

This section provides complementary analyses that further contextualize the main experimental results and clarify several practical design choices for vision–tabular multi-modal learning. We focus on three key axes that most directly affect model performance, generalization, and robustness: (i) fusion architecture, (ii) training strategy, and (iii) backbone selection for each individual modality. Across diverse datasets and task types, we highlight consistent empirical patterns and common failure modes, and distill actionable implications for building more reliable, robust, and effective vision–tabular models.

**Late fusion shows consistent robustness across datasets and tasks.** As shown in Figure 7 and Table 2, late fusion methods (Concat and MAX) consistently rank among the top performers across datasets and task types, whereas deep fusion models fail to show stable advantages. Moreover, vision–language models pre-trained on vision–text data also underperform on vision–tabular tasks, indicating limited transferability. **These results suggest that specialized vision–tabular models remain necessary and that late fusion should be prioritized for robustness.**

MUL is an exception, as it suffers from numerical instability in regression tasks due to scale sensitivity. Specifically, MUL performs channel-wise multiplicative fusion, where large activations or mismatched feature scales across modalities can be amplified through the product operation. This issue is especially problematic for regression, where outputs are unconstrained continuous values and extreme fused representations can lead to very large errors. We observe this behavior across multiple regression datasets, whereas such severe instability does not appear in classification tasks. By contrast, other late-fusion methods such as Concat and MAX remain much more stable overall. Therefore, apart from MUL, simple fusion strategies remain reliable across settings.

**Contrastive pretraining does not yield consistent gains across datasets.** Figure 6 compares MMCL with its ResNet-50 backbone. MMCL exhibits setting-dependent behavior, improving some datasets while degrading others, which indicates that contrastive alignment alone is insufficient for vision–tabular tasks. This is largely because image and table modalities often describe different semantic aspects (e.g., demographics vs. local pathology). In such cases, enforcing strict alignment can suppress modality-specific features that are more informative for downstream tasks. **Future designs should therefore prioritize preserving modality-specific discriminative features over strict semantic alignment.**

**Backbone capacity bounds the headroom of multi-modal fusion.** Most vision–tabular models adopt standard backbones (ResNet-50 for images and lightweight Transformers for tables), yet our results show these choices are suboptimal. On most tasks, ViT-16 consistently outperforms ResNet-50, while TabPFN-v2 substantially surpasses Transformers trained from scratch. Since multi-modal performance is bounded by unimodal representation quality, backbone capacity defines the

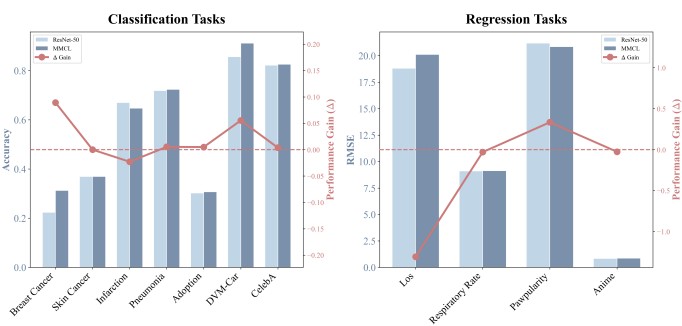

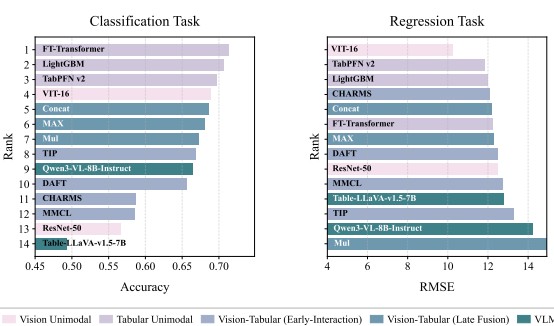

*Figure 6.* Performance comparison between MMCL and its visual backbone (ResNet-50) across classification and regression tasks. The bar charts display the performance metrics (Accuracy for classification; RMSE for regression), while the red line plots the performance gain ($\Delta$) of MMCL over the baseline.

*Figure 7.* Rank results on discriminative prediction tasks. Accuracy denotes the mean accuracy averaged over all classification datasets, while RMSE denotes the mean root mean squared error averaged over all regression datasets.

performance ceiling of fusion models. **Thus, backbone selection should be treated as a core design factor, alongside fusion mechanisms, to raise the overall performance upper bound.**

## C. Challenges and Limitations in Vision–Tabular Multi-modal Learning

### C.1. Discriminative Prediction Task

**Challenges.** Unlike vision-text tasks, where both modalities are typically embedded in high-dimensional continuous spaces and often describe the same underlying semantics, vision-tabular learning combines two highly heterogeneous yet complementary sources of information. Images provide dense, continuous spatial and appearance cues, whereas tables encode discrete, structured attributes with limited direct correspondence to specific visual regions. Such heterogeneity makes it inherently challenging to learn discriminative and separable representations for each modality while still enabling reliable fusion within a shared embedding space. It also weakens the applicability of methods transferred from the vision-text domain (e.g., contrastive learning), whose assumptions about semantic overlap and cross-modal alignment often do not hold in vision-tabular settings.

**Limitations.** We identify three primary limitations: 1) Alignment mismatch: Many existing vision-tabular approaches overemphasize language-style alignment objectives. However, when semantic granularity differs across modalities, enforcing strict alignment or deep coupling can over-regularize the model and compress modality-specific discriminative cues, resulting in optimization instability and weaker generalization. 2) Representation conflict: Current learning strategies often fail to preserve the distinctiveness of heterogeneous evidence within a joint embedding space. This deficiency manifests as negative transfer, where introducing an additional modality interferes with decision-relevant representations and degrades predictive performance. 3) Suboptimal backbones: Encoder/backbone choices are typically made by convention rather than through systematic evaluation tailored to heterogeneous fusion. This limits representation quality, constrains the attainable benefits of multi-modal integration, and may further exacerbate interference effects.

### C.2. Generative Reasoning Task

**Challenges.** Vision-tabular generative reasoning imposes three tightly coupled capability requirements: (i) robust cross-modal perception under long, mixed contexts that interleave text, serialized tables, and images; (ii) structure-aware grounding that can precisely localize relevant rows/columns and retrieve cell-level values; and (iii) constrained computation and operator execution for numerical and logical reasoning. A key challenge is that table serialization substantially expands the context and increases verbosity, which dilutes attention to key evidence and destabilizes cross-modal evidence selection. Moreover, tables encode not only fine-grained values but also explicit row-column structure, so successful reasoning requires coordinate-like localization, retrieval, and filtering rather than relying on linear text semantics alone. Finally, many vision-tabular questions can be formalized as operator chains (e.g., comparison, filtering, aggregation, and computation), which require faithful intermediate state tracking and reliable execution of tightly constrained arithmetic/statistical operations over long contexts. In high-constraint domains such as medicine, these demands are further amplified by the need to generate and execute complex programs (e.g., SQL) involving multi-table joins, nested predicates, and fine-grained temporal constraints.

*Table 4.* Robustness of MCR to reference choices and training-data size stability. Red indicate modalities exhibiting negative transfer.

*(a)* Robustness to alternative non-informative references. MCR is computed using zero, min, and median embeddings as references.

| Reference | Model | Adoption | | CelebA | | LOS | |
|---|---|---|---|---|---|---|---|
| | | Img | Tab | Img | Tab | Img | Tab |
| Zero | TIP | 2.96 | 97.04 | -94.10 | 5.90 | 91.70 | 8.30 |
| | Concat | -14.71 | 85.29 | 62.59 | 37.41 | 95.71 | 4.29 |
| | MAX | 84.47 | 15.53 | 55.30 | 44.70 | 78.34 | 21.66 |
| | DAFT | 57.73 | 42.27 | 68.84 | 31.16 | 95.69 | 4.31 |
| Min | TIP | 2.96 | 97.04 | -94.10 | 5.90 | 91.70 | 8.30 |
| | Concat | -5.24 | 94.76 | 56.93 | 43.07 | 3.58 | 96.42 |
| | MAX | 84.47 | 15.53 | 55.30 | 44.70 | 78.34 | 21.66 |
| | DAFT | 58.80 | 41.20 | 68.20 | 31.80 | 11.66 | 88.34 |
| Median | TIP | 6.43 | 93.57 | -94.10 | 5.90 | 71.02 | 28.98 |
| | Concat | 32.22 | 67.78 | 62.59 | 37.41 | 68.05 | 31.90 |
| | MAX | 23.81 | 76.19 | 55.30 | 44.70 | 61.16 | 38.84 |
| | DAFT | -14.81 | 85.19 | 67.11 | 32.89 | -0.02 | 99.98 |

*(b)* Robustness to training-data size. The mean-embedding reference is estimated from 75%, 50%, and 25% of the training data.

| Reference | Model | Adoption | | CelebA | | LOS | |
|---|---|---|---|---|---|---|---|
| | | Img | Tab | Img | Tab | Img | Tab |
| Mean (75%) | TIP | 0.00 | 100.00 | -0.64 | 99.36 | 14.30 | 85.70 |
| | Concat | -15.58 | 84.42 | 61.70 | 38.30 | -13.32 | 86.68 |
| | MAX | 0.00 | 100.00 | 69.06 | 30.94 | 18.61 | 81.39 |
| | DAFT | -80.25 | 19.75 | 50.72 | 49.28 | -0.45 | 99.55 |
| Mean (50%) | TIP | 0.00 | 100.00 | -0.64 | 99.36 | 15.36 | 84.64 |
| | Concat | -16.88 | 83.12 | 61.40 | 38.60 | -14.36 | 85.64 |
| | MAX | 0.00 | 100.00 | 69.22 | 30.78 | 18.59 | 81.41 |
| | DAFT | -80.25 | 19.75 | 50.49 | 49.51 | -0.37 | 99.63 |
| Mean (25%) | TIP | 0.00 | 100.00 | -0.64 | 99.36 | 18.10 | 81.90 |
| | Concat | -14.29 | 85.71 | 61.75 | 38.25 | -12.13 | 87.87 |
| | MAX | -2.04 | 97.96 | 68.88 | 31.12 | 18.50 | 81.50 |
| | DAFT | -80.25 | 19.75 | 50.65 | 49.35 | -0.11 | 99.89 |

**Limitations.** Despite strong performance from proprietary state-of-the-art systems, general-purpose VLMs still exhibit systematic weaknesses in this setting. First, the perception of visual evidence and table structure remains insufficient. Performance degrades as tables grow larger, suggesting that increased context length reduces attention to critical visual cues and leads to less stable predictions. In addition, open-source models show pronounced deficits in table-structure grounding. Accuracy drops as tasks demand progressively more precise grounding, moving from identifying the target entity to localizing the correct row and finally retrieving cell-level attributes. This trend indicates limited structure-aware representations and inadequate control over intermediate retrieval states. Second, constrained statistical computation and program generation remain weak. Models struggle with tightly constrained counting and conditional statistics over large tables, implying that prompt-only reasoning over rendered or serialized tables is insufficient. In clinically realistic settings (e.g., EHRXQA), open-source models additionally fail to robustly translate questions into executable SQL and remain limited even when correct evidence is provided, reflecting a persistent gap in the end-to-end "programmatic retrieval + expert reasoning" loop, exacerbated by the nested logic and fine-grained temporal constraints that are common in clinical queries.

## C.3. Recommendation

Future research should prioritize enhancing the reliability and robustness of evidence utilization. For discriminative prediction tasks, designs must address heterogeneous discriminative representation learning and stable fusion. This involves learning distinct representations for each modality within a shared embedding space and employing selective fusion mechanisms to prevent negative transfer. Additionally, backbone selection should be evaluated systematically, prioritizing stronger encoders to avoid premature performance ceilings. For vision-tabular generative reasoning under strict numerical constraints, future work should investigate executable table representations with explicit structure and indexing, together with tool-augmented operators for reliable filtering, aggregation, and computation with execution-based checking. Finally, for clinical QA, a promising direction is neuro-symbolic system design that integrates schema-aware program generation with explicit constraints and execution-time verification to improve robustness across stages. Beyond these directions, future work may also consider adaptive model routing, efficient initialization and knowledge inheritance, and open-environment adaptation to further improve the scalability and robustness of vision-tabular multi-modal learning (Ma et al., 2026; Wang et al., 2023; Xia et al., 2024; Lai et al., 2026).

## D. Robustness Analysis of Modality-Specific Diagnostic Metrics

**Robustness of MCR to reference choices.** In the main experiments, MCR replaces one modality with the mean embedding computed over the training set. To validate whether our conclusions are driven by this specific reference choice, we additionally test three alternative non-informative references: zero embedding, min embedding, and median embedding. The zero embedding is an all-zero vector; the min embedding is a constant vector filled with the global minimum scalar of the

*Table 5.* Robustness of MIR under alternative unimodal-performance proxy definitions. Top-2 averages the two best unimodal errors, Avg-all averages all unimodal errors, and Median uses the median unimodal error.

| Proxy | Breast Cancer | Skin Cancer | Infarction | Pneumonia | Adoption | DVM-Car | CelebA | LOS | Respiratory Rate | Pawpularity | Anime |
|---|---|---|---|---|---|---|---|---|---|---|---|
| Top-2 | 1.32 | 1.77 | 1.01 | 1.00 | 1.22 | 1.11 | 0.99 | 1.04 | 0.78 | 1.02 | 1.75 |
| Avg-all | 1.29 | 1.70 | 0.96 | 0.99 | 1.20 | 1.09 | 0.99 | 0.99 | 0.69 | 1.01 | 1.70 |
| Median | 1.27 | 1.76 | 0.89 | 1.00 | 1.22 | 1.09 | 0.99 | 0.92 | 0.73 | 1.02 | 1.72 |

training embedding distribution; and the median embedding is a constant vector filled with the global median scalar. We conduct this analysis on three representative datasets, Adoption, CelebA, and LOS, and four representative multi-modal models, TIP, Concat, MAX, and DAFT. As shown in Table 4a, although the exact MCR magnitudes vary across reference choices, the main qualitative conclusion remains unchanged: negative transfer still appears under all tested reference choices. Quantitatively, the average sign agreement between the original mean-embedding MCR and these alternative references is 63.4%, suggesting that the observed pattern is not solely an artifact of the mean-embedding reference.

**Robustness of MCR to training-data size.** We also examine whether the mean-embedding reference is sensitive to the amount of training data used to estimate it. Specifically, we recompute the mean embedding using reduced training subsets with 75%, 50%, and 25% of the original training data, and evaluate MCR on the same three representative datasets and four models. As shown in Table 4b, the resulting MCR signs are highly stable across different training-data sizes, with 100.0% sign agreement relative to the full-data mean embedding and an average Spearman correlation of 0.99. This indicates that the MCR analysis is robust to the amount of training data used to estimate the mean-embedding reference in our tested settings.

**Robustness of MIR to proxy definitions.** MIR uses the error of the best-performing unimodal baseline as a proxy for dataset-level modality informativeness. To examine whether MIR is sensitive to this specific Top-1 proxy definition, we additionally consider three alternative proxies: Top-2, which averages the errors of the two best unimodal models; Avg-all, which averages the errors of all unimodal models; and Median, which uses the median unimodal error. As shown in Table 5, the alternative MIR definitions preserve the overall modality-direction trends. Across these alternatives, the average direction agreement with the original Top-1 MIR, measured by whether MIR is larger or smaller than 1, is 78.8%, and the average Spearman correlation is 0.694. These results suggest that MIR is reasonably robust to the proxy definition and is not driven by the specific Top-1 formulation.

Overall, these additional analyses suggest that the qualitative conclusions drawn from MCR and MIR are reasonably stable under alternative reference and proxy choices. At the same time, MCR and MIR should be viewed as empirical diagnostic metrics rather than theoretically grounded quantities. Developing a stronger theoretical foundation that connects modality priors, reference choices, fusion behavior, and downstream generalization remains an important direction for future work.

# E. Benchmark Datasets

## E.1. Dataset Modality Evidence Analysis

We analyze the modality evidence required by the benchmark datasets. For discriminative prediction tasks, there is no explicit oracle rule that specifies whether the image or tabular modality is strictly necessary for each label, since the target is determined by statistical dependence rather than by question answering. We therefore manually inspect the image and tabular features of each dataset and find that both modalities provide meaningful evidence in most cases. For example, in the Adoption dataset, pet appearance from the image and attributes such as health condition and adoption fee from the table are both relevant to the final prediction.

For generative reasoning tasks, the evidence source can be more explicitly identified from the question design. We categorize each question according to the modality of evidence required for answering, including image-only, table-only, text-only, and their combinations. As shown in Figure 8, the reasoning benchmarks contain a diverse mixture of question types, covering both single-modality and multi-modality evidence requirements. This breakdown further supports the suitability of our dataset selection. The selected datasets are not dominated by questions answerable from a single modality alone; instead, most questions require integrating evidence from multiple modalities.

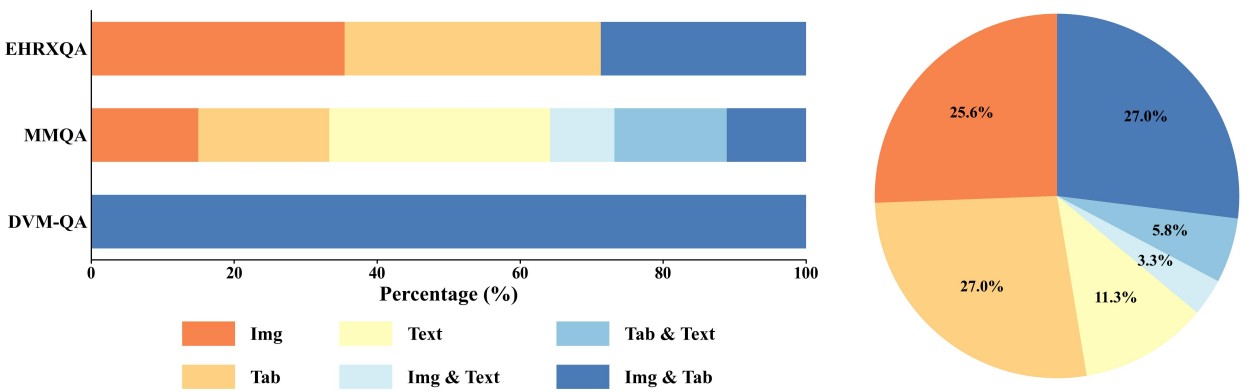

*Figure 8.* Distribution of question types in the generative reasoning benchmarks, grouped by the modality of evidence required for answering: image-only, table-only, text-only (available only in MMQA), and their combinations. The left panel presents the distribution for each individual subset, while the right panel summarizes the overall distribution across all reasoning data.

### E.2. Discriminative Prediction Datasets

This section summarizes the discriminative prediction datasets used in VT-Bench, including their sources, basic statistics, and data splits. We also describe the construction process and prediction targets of three newly built datasets. Figure 9 shows prompt template used for fine-tuning Qwen on the skin-cancer diagnostic task as an example.

#### E.2.1. PUBLIC DATASETS

We introduce eight public datasets used in our benchmark. For all datasets, the reported sample sizes are computed after removing instances with missing values. All images are resized to a resolution of $224 \times 224$ for consistency across datasets.

**Breast Cancer.** Breast Cancer is a curated subset of the Curated Breast Imaging Subset of DDSM (CBIS-DDSM), comprising 3,287 mammogram images. Each sample contains mammographic image data with accompanying region-of-interest segmentation and pathology-related annotations/metadata. The dataset is derived from the Digital Database for Screening Mammography (DDSM) and collected from multiple clinical sites for CADx/CADe research. Each instance corresponds to a mammography case, and the cases are labeled into normal, benign, or malignant categories. The task is to classify mammograms according to breast cancer status based on image information. We split the dataset into training/validation/test sets using an 8:1:1 ratio. The dataset is available at https://www.kaggle.com/datasets/awsaf49/cbis-ddsm-breast-cancer-image-dataset.

**Skin Cancer.** The Skin Cancer dataset (PAD-UFES-20) contains 2,298 labeled skin images, including clinical/dermoscopic photographs, and is accompanied by patient-level metadata (e.g., age, gender, anatomical site, and clinical history) provided in tabular form. Each sample represents a skin lesion case with a diagnostic label covering multiple lesion categories (e.g., melanoma, basal cell carcinoma, and other diagnostic groups). The objective is to distinguish different skin lesion types/skin cancers using image and metadata. We also use the same division for the dataset. The dataset is publicly accessible at https://www.kaggle.com/datasets/mahdavi1202/skin-cancer.

**Infarction.** We use the UK Biobank(Bycroft et al., 2018) myocardial infarction (Infarction) dataset, which contains 36,167 image–tabular pairs. Each sample includes 74 disease-related tabular features, consisting of 26 categorical variables (e.g., alcohol drinking status) and 48 continuous variables (e.g., average heart rate), together with corresponding cardiac imaging data. Each instance represents an individual participant, and the label indicates whether myocardial infarction is present (binary classification). Following the preprocessing protocol in TIP(Du et al., 2024), we conduct data preparation accordingly. Note that UK Biobank releases may differ across official updates and access versions, which can introduce distribution shifts compared with earlier implementations. The dataset is accessible through the UK Biobank application portal at https://www.ukbiobank.ac.uk/enable-your-research/apply-for-access.

---

**Skin Cancer Diagnostic Classification Prompt Template**

**Task Overview**

You are tasked with predicting the **diagnostic** (diagnosis category) for each skin lesion based on both the tabular features and the image data provided in the skin cancer dataset. The **diagnostic** variable indicates the type of skin lesion/disease and is a categorical variable with the following possible values:

- **BCC**: Basal Cell Carcinoma - The most common form of skin cancer, usually slow-growing and rarely metastasizes
- **SCC**: Squamous Cell Carcinoma - The second most common skin cancer, can be more aggressive than BCC
- **MEL**: Melanoma - The most dangerous form of skin cancer, can spread to other parts of the body if not caught early
- **NEV**: Nevus - Benign mole or birthmark, non-cancerous
- **ACK**: Actinic Keratosis - Pre-cancerous skin lesion caused by sun damage, can develop into SCC
- **SEK**: Seborrheic Keratosis - Benign skin growth, non-cancerous, also known as seborrheic wart

You need to make a prediction for **diagnostic** using both the patient's **tabular information** (such as age, gender, lesion characteristics, medical history, etc.) and its **skin lesion image**. The output should be one of the following: **BCC**, **SCC**, **MEL**, **NEV**, **ACK**, or **SEK**, representing the predicted diagnostic category for each skin lesion.

**Dataset Overview**

This skin cancer dataset contains dermatoscopic images of skin lesions along with comprehensive patient and lesion metadata. Each record includes the following information:

- *patient_id*, *lesion_id*, *age*, *gender*, *region*
- *diameter_1*, *diameter_2*, *fitspatrick*, *smoke*, *drink*
- *pesticide*, *background_father*, *background_mother*
- *skin_cancer_history*, *cancer_history*, *has_piped_water*, *has_sewage_system*
- *itch*, *grew*, *hurt*, *changed*, *bleed*, *elevation*, *biopsed*
- *diagnostic* (target), *img_id* (image file identifier)

**Important Information**

- **Diagnostic** is a categorical variable representing the type of skin lesion/cancer.
- Features such as *age*, *gender*, *region*, *fitspatrick*, *skin_cancer_history*, lesion characteristics (*itch*, *grew*, *hurt*, *changed*, *bleed*, *elevation*), *diameter*, and the **skin lesion image** itself are important.
- *Region* indicates where on the body the lesion is located, which can be important as some types of skin cancer are more common in sun-exposed areas.
- *fitspatrick* relates to skin phototype and UV-response.
- Lesion characteristics are key clinical indicators for differential diagnosis.
- The dermatoscopic image (*img_id*) should be analyzed together with the tabular metadata.

**Answer Format**

For each sample respond with: **answer:**[*PredictionResult*]
where *PredictionResult* is one of **BCC**, **SCC**, **MEL**, **NEV**, **ACK**, or **SEK**.

Example outputs:
- answer:BCC
- answer:NEV
- answer:MEL

*Figure 9.* Prompt template used for fine-tuning Qwen on the skin-cancer diagnostic task. The template instructs the model to predict the diagnostic category (BCC, SCC, MEL, NEV, ACK, or SEK) based on both tabular patient/lesion metadata and the corresponding dermatoscopic image.

**Adoption.** The PetFinder adoption prediction dataset contains 14,652 pet profiles with structured metadata. For feature construction, we convert the raw long-text description into statistical features and sentiment-analysis features, yielding a final 25-dimensional tabular representation for each sample, including 24 numerical variables and 1 categorical variable, which describe the pet's attributes and status. Each instance corresponds to an online pet profile, and the label is AdoptionSpeed (five classes, from 0 to 4). The task is to predict adoption speed from profile metadata (five-class classification). We split the dataset into training/validation/test sets using an 8:1:1 ratio. The dataset is available at https://www.kaggle.com/competitions/petfinder-adoption-prediction.

**CelebA.**   CelebA (CelebFaces Attributes) is a large-scale face attribute dataset containing 202,599 celebrity face images from 10,177 identities. Each image is annotated with 39 binary facial attributes (e.g., Big Nose). Each sample represents a celebrity face image with attribute-level annotations, and we use Attractive as the target label, forming a binary classification task. We adopt the same 8:1:1 split for training/validation/test. The dataset is available at https://www.kaggle.com/datasets/jessicali9530/celeba-dataset.

**DVM-Car.**   The DVM-CAR dataset is a large-scale automotive multi-modal dataset containing 176,414 cleaned car images and corresponding tabular records for 899 car models sold in the UK market over the past 20 years. Each sample consists of image data (uniformly resized with background removed) and non-visual attributes provided in six CSV tables, covering car specifications and market information (e.g., brand, price, and sales). The dataset is collected and released by the Deep Visual Marketing (DVM) project for business-related research in the automotive industry. Each instance corresponds to a specific car model (with associated images and metadata). We adopt the same preprocessing protocol as CHARMS (Jiang et al., 2024) and use this dataset for multi-modal representation learning on automotive analytics tasks. The dataset is available at https://deepvisualmarketing.github.io/.

**Pawpularity.**   The PetFinder Pawpularity dataset contains 7,930 pet profiles, where each sample consists of a pet image and associated structured metadata. The tabular modality includes 8 binary indicators describing photo characteristics (e.g., whether the pet face is visible and frontal), which are treated as categorical features. Each instance is annotated with a Pawpularity Score, and the task is to predict the popularity score from multi-modal inputs. We split the dataset into training/validation/test sets using an 8:1:1 ratio. The dataset is available at https://www.kaggle.com/competitions/petfinder-pawpularity-score.

**Anime.**   The Anime dataset contains 12,513 anime entries with structured content information, together with auxiliary files that support large-scale analysis of user preferences. Each sample is represented by tabular attributes such as title, genre, demographics, and popularity-related statistics, and may additionally include user-interaction records (e.g., user scores) for preference modeling. The dataset is collected to facilitate research on anime content characteristics and viewer preference analysis. Each instance corresponds to an anime item, and the target label is the overall score assigned to the anime. We split the dataset into training/validation/test sets using an 8:1:1 ratio. The dataset is available at https://www.kaggle.com/datasets/dbdmobile/myanimelist-dataset.

### E.2.2. Constructed Datasets

In our review, we found that high-quality, large-scale medical vision–tabular prediction datasets remain limited, and existing benchmarks are often dominated by classification tasks. This makes it difficult to systematically evaluate vision-tabular multi-modal methods under realistic clinical settings. To address this gap, we construct three large-scale vision–tabular prediction tasks based on MIMIC-IV v2.2(Johnson et al., 2023) and MIMIC-CXR v2.0.0(Johnson et al., 2019), extending the coverage of VT-Bench in the medical domain.

For dataset construction, we align each chest X-ray study with a specific hospital admission using timestamps, requiring the imaging time to fall strictly within the corresponding hospitalization period. This ensures temporal synchronization between unstructured imaging data and structured clinical records. We then apply strict quality control by removing any sample with missing tabular modality, so that all downstream tasks have complete multi-modal inputs. The resulting tabular features include three groups of key clinical variables: demographics (e.g., age, sex), vital signs (e.g., heart rate, temperature, SpO2), and laboratory tests (e.g., WBC, hemoglobin, platelet count).

To cover multiple clinical decision stages, we design one classification task and two regression tasks, spanning diagnostic and prognosis-related scenarios. For all three tasks, we split the data into training/validation/test sets with an 8:1:1 ratio. To prevent subject-level leakage, we perform the split at the patient level rather than the sample level, ensuring that all records from the same patient appear in only one subset. We briefly describe the three newly constructed prediction datasets below.

**Pneumonia.**   This task focuses on pneumonia diagnosis. The model must integrate radiographic findings from chest X-rays (e.g., infiltrates) with inflammation-related indicators from the tabular modality, reflecting how clinical diagnosis relies on complementary evidence from different sources.

**Length of Stay (LOS).**    This task predicts the length of hospital stay, which is relevant to practical needs such as capacity planning and resource allocation. The model is expected to combine imaging cues related to disease severity with tabular information from vital signs, providing a more complete basis for prognosis-related assessment.

**Respiratory Rate.**    This task predicts respiratory rate. It emphasizes linking pulmonary imaging patterns (e.g., consolidation, pleural effusion) to corresponding physiological signals, and is intended to evaluate whether the model captures clinically meaningful cues associated with respiratory function.

### E.3. Generative Reasoning Datasets

#### E.3.1. EHRXQA.

EHRXQA (Bae et al., 2023) is a multi-modal medical question answering benchmark designed for the setting where a chest X-ray is paired with structured electronic health records (EHRs). Instead of providing explicit vision-tabular evidence pairs, relevant patient information is stored in a relational database, so answering a question typically requires database retrieval followed by cross-modal clinical reasoning. This design reflects common clinical workflows and enables evaluation of both evidence acquisition and answer generation.

In VT-Bench, we evaluate EHRXQA using a two-stage pipeline and report end-to-end accuracy. In Stage 1, the model generates a structured execution plan that specifies whether retrieval is needed and, when applicable, produces an SQL query. The query is executed on the benchmark database to retrieve patient sub-tables and the associated chest X-ray(s). In Stage 2, the model is provided with the question together with the retrieved evidence and execution plan, and then generates the final answer. The prompt templates used in this evaluation are shown in Figure 10.

To better distinguish retrieval failures from reasoning errors, we further conduct a decoupled evaluation. Stage 1 is evaluated based on SQL executability, since EHRXQA does not release standardized retrieval code or official retrieval outputs. For Stage 2, we measure question answering performance under the assumption that the required evidence is correctly retrieved. Specifically, we automatically rewrite the NeuralSQL annotations released with EHRXQA into executable SQL queries that return the necessary patient information for each question, and we verify the resulting tables to ensure they contain all information needed to answer the question. The model is then conditioned on these verified retrieval results and evaluated on its answer accuracy, providing a direct estimate of its clinical reasoning ability given correct evidence.

#### E.3.2. MULTI-MODALQA.

Multi-ModalQA (MMQA) (Talmor et al., 2021) is a multi-modal question answering benchmark that requires jointly using evidence from text, tables, and images. Unlike single-source QA, MMQA often requires aligning entities and attributes across modalities and performing multi-step reasoning to derive the final answer, making it suitable for evaluating cross-modal aggregation and reasoning.

MMQA defines four unimodal sub-tasks: *TextQ*, *TableQ*, *ImageQ*, and *ImageListQ*, indicating that each question is answerable using evidence from the corresponding modality only. It further summarizes multi-step reasoning into three operations and accordingly specifies 12 compositional sub-task types:

- **Compose(A,B).** Chained reasoning where the model first answers sub-question *B* and then uses its output to solve *A*. The corresponding sub-tasks are `Compose(TextQ,TableQ)`, `Compose(TableQ,ImageListQ)`, `Compose(ImageQ,TableQ)`, `Compose(TableQ,TextQ)`, `Compose(TextQ,ImageListQ)`, and `Compose(ImageQ,TextQ)`.
- **Intersect(A,B).** Set-based reasoning where the model derives candidate answer sets from two evidence sources and returns their intersection. The corresponding sub-tasks are `Intersect(TableQ,TextQ)`, `Intersect(ImageListQ,TableQ)`, and `Intersect(ImageListQ,TextQ)`.
- **Compare(A,B).** Comparative reasoning that involves numerical or temporal table fields; the model aligns entities and selects the valid result by comparison. The corresponding sub-tasks are `Compare(Compose(TableQ,ImageQ),TableQ)`, `Compare(TableQ,Compose(TableQ,TextQ))`, and `Compare(Compose(TableQ,ImageQ),Compose(TableQ,TextQ))`.

For evaluation, we follow the original evaluation protocol and report end-to-end accuracy as the overall metric. The prompt template adopted in this evaluation setting is shown in Figure 11.

### Stage 1: Strategic Planning and SQL Generation

You are a clinical question answering assistant for the EHRXQA dataset.

Database: `{db_path}`

Schema: `{DB_SCHEMA_TEXT}`

**Execution environment**

You cannot execute SQL. You only write SQL queries as plain text. An external Python runner will execute the queries and return results in the next turn.

**Image retrieval** Images are stored under `{image_root}` with:

`{image_root}/{pXX}/{pXXXXXX}/{sYYYYYY}/{image_id}.jpg`.

If an image is needed, the SQL must select: `subject_id, study_id, image_id`.

**Temporal constraints (mandatory)** MIMIC-IV and MIMIC-CXR use de-identified time shifting.

Assume: `current_time = 2105-12-31 23:59:00`. Never use `CURRENT_TIMESTAMP` or `NOW()`.

Example:

```
charttime >= DATETIME("2105-12-31 23:59:00", "-24 hours")
```

**Decision rules**

- `need_sql`: True iff patient-specific evidence is required.
- `need_image`: True iff visual inspection of CXR is required.

**Output (JSON)**

```
{ need_sql, sql_queries, need_image, reasoning, final_answer }
```

### Stage 2: Multi-modal Clinical Reasoning

You previously generated a plan for:

**Question:** `{question}`

**Plan:** `{stage1_plan_json}`

The external system executed the SQL and returned: `{sql_text}`

The corresponding CXR images for the selected rows are attached.

Based only on (1) question, (2) plan, (3) SQL results and images, provide the final answer.

**Output**

```
{ "final_answer": "only the answer content", "explanation": "brief reasoning trace" }
```

**Constraint:** `"final_answer"` must be only the answer itself (e.g., `"low lung volumes"`).

*Figure 10.* The two-stage prompt design for EHRXQA evaluation. Stage 1 acquires structured evidence via SQL under strict temporal constraints. Stage 2 integrates returned tabular evidence and CXR images for clinical reasoning.

### E.3.3. DVM-Car QA.

Existing benchmarks for vision–tabular reasoning remain limited and do not provide a systematic evaluation of several core capabilities: cross-modal grounding between an image and a table, understanding table structure (e.g., locating the correct row or attribute), and performing numerical reasoning under logical constraints. While the latter two are commonly studied in purely tabular settings, they are not well characterized in vision–tabular scenarios where cross-modal grounding and tabular reasoning must be combined. To address this gap, we build DVM-Car QA on top of the DVM-Car prediction dataset(Huang et al., 2022) and extend it into a fine-grained benchmark for joint reasoning over an image and a table (see Figure 12 for the construction pipeline).

Each instance in DVM-Car QA is represented as a triplet $(I, T, q)$. Here, $I$ is a front-view car image, and $T$ is a structured

---

**Multi-ModalQA Prompt**

You are an expert multi-modal AI assistant. Please answer the following question based *only* on the provided context (texts, table, and images).—

— **START IMAGE CONTEXT** —
{image_1}, {image_2}, ...
— **END IMAGE CONTEXT** —

— **START TEXT CONTEXT** —
Text Snippet 1 (Title: Kirov Stadium): {text_1}

Text Snippet 2 (Title: Kievan Rus'): {text_2}
....
— **END TEXT CONTEXT** —

— **START TABLE CONTEXT** —
{markdown_table}
— **END TABLE CONTEXT** —

— **QUESTION** —
{question}

---

— **ANSWER INSTRUCTIONS** —
Provide only the final answer to the question, with no explanation, reasoning, or conversational text. For example, if the answer is 'Mask', just output 'Mask'.

*Figure 11.* The multi-modal reasoning prompt for MMQA. This template strictly enforces zero-shot constraints and concise output across text, tabular, and visual modalities.

table that contains $n$ candidate vehicles ($n \in 10, 20, 50$) described by 15 fixed attributes. Importantly, exactly one row in $T$ uniquely corresponds to the vehicle shown in $I$, which we refer to as the target row. The question $q$ is written in natural language and is defined with respect to the target vehicle and/or the other candidates in the table. We denote the visual cues that allow the target row to be uniquely identified as *Vision Alignment Keys*. Our task formulation ensures that answering each question strictly requires both modalities: the model must first extract visual cues to locate the target row in the table, and then derive the final answer solely from the table. For example, in Figure 12, the image may expose multiple cues such as maker and color, yet only color yields a unique match because the table contains exactly one red vehicle. Accordingly, a model must determine the effective alignment cues, perform row localization, and then proceed to attribute reading or constrained aggregation. To control matching difficulty and ambiguity, we provide three alignment settings: Color only, Maker only, and Color & Maker.

Given the same input pair $(I, T)$, we design four sub-tasks with increasing complexity: (i) *Row Localization*, which identifies the target row using visual cues; (ii) *Cell Retrieval*, which outputs a specified attribute value of the target vehicle; (iii) *Constrained Counting*, which filters the sub-table based on conditions defined relative to the target vehicle and counts the remaining entries; and (iv) *Conditional Mean*, which computes the mean of a numerical attribute over the filtered set. These tasks progressively raise the requirements on cross-modal grounding, table-structure understanding, and numerical reasoning. In addition, by varying the *Vision Alignment Keys* and the sub-table size, we systematically control ambiguity and context length, enabling an analysis of VLM performance degradation under different perceptual and computational loads. In total, we construct 5,400 samples. The prompt template used in this dataset is shown in Figure 13.

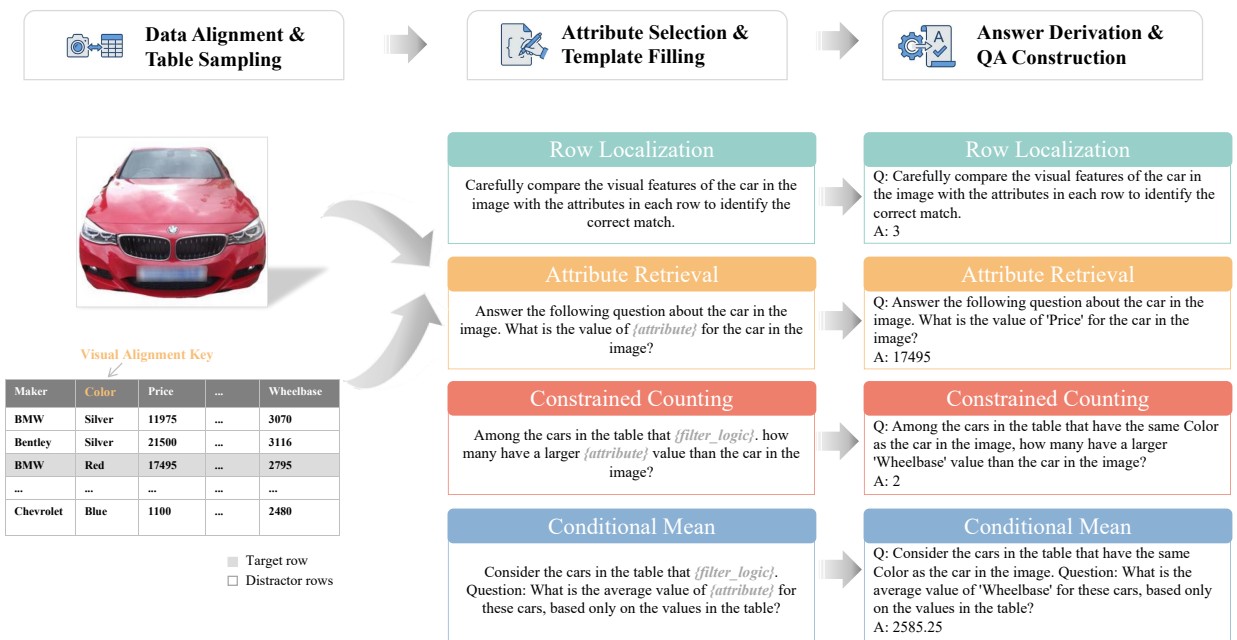

*Figure 12.* **Construction pipeline of DVM-Car QA.** The pipeline includes three stages: (1) **Data alignment & sampling**, where each car image is matched to its table row and distractor rows are added so the target row is uniquely identifiable by a visual alignment key (e.g., `Color=Red` in the example); (2) **Attribute selection & template instantiation**, where attributes and target-relative constraints are sampled to generate questions; (3) **Answer derivation**, where ground-truth answers are programmatically computed, forming four task types: row localization, attribute retrieval, constrained counting, and conditional mean.

# F. Benchmark Models

In this section, we provide introductions to tree-based and deep learning models and hyperparameter grids. We implement adaptive hyperparameter optimization based on the Optuna framework and following previous studies (Liu et al., 2024), fixing the batch size at 1024 and conducting 100 independent trials through train-validation splits to prevent test set leakage, with the best-performed hyperparameters fixed during the final 15 seeds. The hyperparameter grids are shown in Table 6.

## F.1. Vision Unimodal Models

**ResNet-50**  ResNet-50(He et al., 2016) is a 50-layer instantiation of deep residual learning, which introduces identity skip connections and formulates stacked nonlinear layers as learning residual functions with respect to their inputs to alleviate optimization difficulties in deep networks. This design makes substantially deeper networks easier to train and improves accuracy by increasing depth while keeping complexity manageable, and it has become a widely adopted backbone for large-scale visual recognition as well as downstream detection and segmentation systems.

**ViT-B/16**  ViT-B/16(Dosovitskiy et al., 2021) is a pure Transformer architecture for vision that represents an image as a sequence of fixed-size patches (16×16) and applies standard self-attention directly over these patch tokens, without relying on convolutional backbones. With large-scale pre-training and subsequent transfer, ViT achieves strong image classification performance across benchmarks such as ImageNet, CIFAR-100, and VTAB, while being more compute-efficient to train than competitive convolutional networks.

## F.2. Tabular Unimodal Models

**LightGBM**  LightGBM(Ke et al., 2017) is an efficient Gradient Boosting Decision Tree (GBDT) implementation optimized for large-scale, high-dimensional data. It introduces Gradient-based One-Side Sampling (GOSS) to down-sample low-gradient instances and Exclusive Feature Bundling (EFB) to reduce dimensionality by merging sparse features. These techniques substantially improve training speed and scalability, delivering significant acceleration over conventional GBDT

---

**DVM-Car QA Prompt**

## Table
`{table_content}`

## Image
`{image_content}`

## Question:
You are given a photo of a car and the table above listing several candidate cars. Exactly one row in the table corresponds to the car in the image. Based on the matching row, answer the following question about the car in the image: `{question}`

## OUTPUT FORMAT:
Return a JSON object:
{
    "answer": [VALUE],
    "explanation": "Brief step-by-step reasoning..."
}

---

*Figure 13.* The visual-tabular reasoning prompt for DVM-Car QA. This template evaluates the model's ability to perform cross-modal grounding and attribute retrieval in a zero-shot setting.

models with comparable predictive accuracy.

**TabTransformer** TabTransformer(Huang et al., 2020b) adopts a Transformer-based architecture to contextualize categorical feature embeddings through self-attention, producing more informative and robust representations for tabular learning. It achieves strong supervised performance, often surpassing prior deep learning-based tabular models, while improving robustness to noisy features and enhancing interpretability. Additionally, it supports semi-supervised learning through unsupervised pre-training, yielding consistent accuracy gains.

**TabPFN v2** TabPFN v2(Hollmann et al., 2025) is a tabular foundation model trained on millions of synthetic datasets to perform general-purpose inference for supervised prediction on tables. Designed for small-to-medium datasets (up to 10,000 samples), it frequently outperforms heavily tuned gradient-boosted tree ensembles with significantly lower training overhead. As a generative foundation model, TabPFN v2 further supports fine-tuning and embedding learning, enabling broader workflows like data generation. For datasets with more than 10,000 samples, we subsample to 10,000 instances by using stratified sampling for classification datasets and random sampling for regression datasets, and we report results averaged over three independently sampled subsets with different random seeds.

### F.3. Vision–Tabular Multi-Modal Models

**Concat** Concat(Spasov et al., 2019) propose a parameter-efficient multi-task deep learning model that predicts 3-year MCI→AD conversion by jointly learning AD-versus-healthy classification and integrating structural MRI with demographic, neuropsychological, and APOe4 features. Evaluated on ADNI, it reports strong conversion prediction performance and demonstrates robustness to different preprocessing choices and experimental settings.

**MAX** MAX(Vale-Silva & Rohr, 2021) is a multi-modal deep learning framework for long-term pan-cancer survival prediction that learns modality-specific representations from clinical variables, imaging, and multi-omics data, then fuses them to output conditional survival probabilities over long follow-up horizons. It is designed to operate under missing values or even missing modalities, and was evaluated across 33 cancer types, reporting strong performance against prior survival models under time-dependent metrics while enabling representation-based analyses of cancer heterogeneity.

**MUL** MUL(Duanmu et al., 2020) propose an integrative CNN to predict pathological complete response to neoadjuvant chemotherapy from post-contrast 3D T1-weighted breast MRI together with molecular subtype and demographic variables. Instead of simple feature concatenation, the imaging and non-imaging streams interact via channel-wise multiplicative

fusion, yielding better performance than imaging-only and concatenation baselines.

**DAFT** DAFT(Pölsterl et al., 2021) is a lightweight conditioning module for CNNs that fuses 3D imaging features with tabular clinical variables by dynamically predicting per-channel scale and shift parameters and applying an affine transform to convolutional feature maps. This enables fine-grained image–tabular interaction and improves performance on tasks such as Alzheimer's diagnosis and time-to-dementia prediction compared to standard fusion baselines.

**CHARMS** CHARMS(Jiang et al., 2024) studies cross-modal knowledge transfer from expert-derived tabular attributes to image prediction when tabular inputs are unavailable at inference. It aligns image channels with tabular features via optimal transport and mutual-information maximization, enabling selective transfer of visually relevant numerical and categorical signals, which improves both classification performance and interpretability.

**MMCL** MMCL(Hager et al., 2023) is a multi-modal contrastive pretraining framework that leverages paired imaging and tabular data to learn strong *unimodal* encoders, enabling multi-modal self-supervision during pretraining while supporting unimodal deployment at inference. It combines SimCLR-style image contrast with SCARF-style tabular augmentations, and introduces a simple supervised extension, *label as a feature* (LaaF), by appending the target label as a tabular attribute during multi-modal pretraining.

**TIP** TIP(Du et al., 2024) is a self-supervised tabular–image pre-training framework designed for multi-modal classification under incomplete tabular inputs. It combines masked tabular reconstruction with image–tabular matching and contrastive objectives, together with a tabular encoder tailored to heterogeneous missingness and a multi-modal interaction module, yielding improved performance in both complete and missing-data settings.

### F.4. General-Purpose VLMs

**Table-LLaVA-v1.5-7B** Table-LLaVA-v1.5-7B(Zheng et al., 2024) targets *multi-modal table understanding* by answering table-centric instructions directly from table images, avoiding reliance on serialized formats (e.g., HTML/Markdown). It introduces the large-scale MMTab dataset for training and evaluation, and fine-tunes a LLaVA-based tabular MLLM that substantially outperforms recent open-source multi-modal baselines under both held-in and held-out benchmark settings.

**Qwen3-VL-8B-Instruct** Qwen3-VL-8B-Instruct(Bai et al., 2025) is an instruction-tuned variant of the Qwen3-VL family that supports interleaved text–image–video inputs with a native 256K-token context window. Built on the Qwen3-VL architecture (e.g., interleaved-MRoPE and DeepStack-based multi-level visual feature integration), it targets general-purpose multi-modal understanding and generation under long-context settings.

**Qwen3-VL-8B-Thinking** Qwen3-VL-8B-Thinking(Bai et al., 2025) is a reasoning-oriented variant of Qwen3-VL that retains the same interleaved multi-modal interface and long-context capability (up to 256K tokens), while emphasizing stronger multi-step multi-modal reasoning across single-image, multi-image, and video tasks. It inherits key architectural upgrades such as enhanced interleaved-MRoPE for spatiotemporal modeling, DeepStack for tighter vision–language alignment, and text-based time alignment for more precise temporal grounding in video.

**InternVL3-8B** InternVL3-8B(Zhu et al., 2025) is an open-source multi-modal LLM that performs native multi-modal pre-training, jointly learning vision and language from mixed multi-modal data and pure-text corpora in a single stage. It introduces variable visual position encoding (V2PE) for extended multi-modal context and adopts post-training recipes (e.g., SFT and MPO) plus test-time scaling, achieving strong results across diverse multi-modal benchmarks.

**GLM-4.1V-9B-Thinking** GLM-4.1V-9B-Thinking(Hong et al., 2025) is a reasoning-oriented vision-language model trained with a large-scale multi-modal pre-trained vision backbone and a curriculum-based reinforcement learning recipe (RLCS) to enhance general-purpose multi-modal reasoning. It reports strong results across a broad benchmark suite spanning STEM, video understanding, coding, grounding, and agentic tasks, and is released as an open-source model alongside larger GLM-4.5V and GLM-4.6V variants.

**Llama-3.2-11B-Vision-Instruct** Llama-3.2-11B-Vision (Dubey et al., 2024) is a multi-modal large language model that integrates visual reasoning capabilities into the Llama 3 architecture. It utilizes a separately trained vision adapter consisting

of cross-attention layers to feed image encoder representations into the core Llama 3.1 language model. This 11B parameter model supports high-resolution image understanding and is optimized for tasks such as visual recognition, image reasoning, and captioning, achieving a 128k token context window for sophisticated multi-modal dialogue.

**Pixtral-12B**    Pixtral-12B(Agrawal et al., 2024b) is a 12B vision-language model trained for both natural images and documents, reporting strong multi-modal reasoning and instruction-following performance (e.g., on MMMU) without sacrificing text-only capability. It supports flexible visual tokenization by ingesting images at their native resolution and aspect ratio, and can handle multiple images within a long 128K-token context window for document QA, chart/figure understanding, and general multi-modal reasoning.

**GPT-4.1**    GPT-4.1(OpenAI, 2025) is a flagship large language model optimized for superior coding performance, instruction following, and long-context processing. Released in April 2025, it introduces a 1-million-token context window, enabling the precise handling of massive codebases and complex documents. Engineered for high steerability, GPT-4.1 achieves a 21.4% improvement on benchmarks like SWE-bench over GPT-4o. The model family, comprising Standard, Mini, and Nano variants, provides a scalable solution for balancing latency and cost across diverse production environments.

**Gemini-3-Flash-Preview**    Gemini-3-Flash-Preview(Google DeepMind, 2025) is a lightweight Gemini-family model optimized for low-latency, cost-effective inference while retaining strong general-purpose reasoning and multi-modal capability. It is intended for broad interactive and production workloads, and the accompanying model card summarizes its supported modalities, context capacity, and safety/evaluation characteristics for reproducible use in research and deployment.

### F.5. Tool-Augmented Methods

**StructGPT**    StructGPT((Jiang et al., 2023)) is a tool-augmented framework designed to improve large language models' reasoning ability over structured data. Instead of directly feeding all structured information into the model, StructGPT organizes the reasoning process through an iterative read-and-reason paradigm, where the model interacts with external structured-data interfaces to retrieve relevant evidence and progressively derive the answer. This design makes it suitable for tasks involving tables, databases, and knowledge graphs, where explicit structure-aware access can help reduce irrelevant context and support more reliable reasoning over complex structured inputs.

**Thyme**    Thyme((Zhang et al., 2025)) is a tool-augmented multi-modal reasoning framework that encourages models to think beyond raw image perception by decomposing complex tasks into intermediate reasoning and tool-use steps. It combines visual understanding with external operations such as structured information extraction, programmatic reasoning, or evidence verification, aiming to improve reliability on tasks that require more than direct visual question answering.

## G. Experiment Details and Results

### G.1. Training Details

For discriminative prediction tasks, we follow the official data splits provided in Section E.2.1. All experimental results are reported as the average of three different random seeds to ensure robustness. For generative reasoning tasks, vision–language models (VLMs) are evaluated in a zero-shot setting using their official pretrained checkpoints, without any task-specific fine-tuning. The maximum input context length is set to 1024 tokens, and all models employ their default decoding strategies. We adopt a deterministic inference protocol with fixed prompts, temperature set to 0, and greedy decoding; therefore, there is no analogous seed variability for these tasks, and repeated runs would produce identical outputs under the same protocol. All experiments are implemented in PyTorch and conducted on two NVIDIA A800 GPUs.

### G.2. Discriminative Prediction Task Results

This section presents the detailed performance of unimodal models, vision–tabular multi-modal models, and vision–language models on 11 datasets for discriminative prediction tasks. For classification tasks, AUC and macro-F1 scores are reported in Table 7; for regression tasks, MAE and $R^2$ scores are reported in Table 8. These additional evaluation results complement the accuracy and RMSE metrics reported in the main paper.

*Table 7.* Model performance on Classification Task

*(a)* AUC Results on Classification Datasets

| AUC | Breast Cancer | Skin Cancer | Infarction | Pneumonia | Adoption | DVM-Car | CelebA |
|---|---|---|---|---|---|---|---|
| ResNet-50 | 0.5757 | 0.4655 | 0.5498 | 0.6158 | 0.5719 | 0.9986 | 0.9129 |
| vit-16 | 0.7944 | 0.8049 | 0.6494 | 0.5710 | 0.6331 | 0.9988 | 0.8680 |
| LightGBM | 0.8085 | 0.9633 | 0.7553 | 0.6240 | 0.6638 | 1.0000 | 0.8785 |
| TabPFN v2 | 0.7732 | 0.9714 | 0.7781 | 0.6287 | 0.6823 | 0.8899 | 0.8008 |
| FT-transformer | 0.7691 | 0.9395 | 0.6987 | 0.6113 | 0.6675 | 0.9999 | 0.8753 |
| CHARMS | 0.0190 | 0.6846 | 0.5960 | 0.6330 | 0.5184 | 0.9992 | 0.9034 |
| MMCL | 0.6226 | 0.5038 | 0.7161 | 0.6259 | 0.5604 | 0.9991 | 0.9149 |
| TIP | 0.7322 | 0.9326 | 0.7645 | 0.6083 | 0.5701 | 1.0000 | 0.8882 |
| DAFT | 0.6753 | 0.8861 | 0.7382 | 0.6284 | 0.5359 | 0.9999 | 0.9119 |
| MAX | 0.7564 | 0.9466 | 0.7499 | 0.6432 | 0.6012 | 0.9999 | 0.9150 |
| Concat | 0.7575 | 0.9413 | 0.7454 | 0.5814 | 0.5986 | 0.9999 | 0.9164 |
| Mul | 0.7482 | 0.9363 | 0.7236 | 0.5823 | 0.5348 | 0.9999 | 0.9112 |
| Table-LLaVA-3B | – | – | – | – | – | – | – |
| Qwen2.5-VL-7B-Instruct | – | – | – | – | – | – | – |

*(b)* Macro-F1 Results on Classification Datasets

| Macro-F1 | Breast Cancer | Skin Cancer | Infarction | Pneumonia | Adoption | DVM-Car | CelebA |
|---|---|---|---|---|---|---|---|
| ResNet-50 | 0.1936 | 0.0899 | 0.1142 | 0.0847 | 0.2509 | 0.8218 | 0.8263 |
| vit-16 | 0.5548 | 0.4019 | 0.4907 | 0.4108 | 0.2632 | 0.8437 | 0.7833 |
| LightGBM | 0.5269 | 0.7587 | 0.4736 | 0.5191 | 0.3512 | 0.9777 | 0.7919 |
| TabPFN v2 | 0.3537 | 0.7533 | 0.4867 | 0.4774 | 0.3111 | 0.9959 | 0.8006 |
| FT-transformer | 0.3662 | 0.5138 | 0.5164 | 0.4844 | 0.3011 | 0.9399 | 0.7892 |
| CHARMS | 0.0190 | 0.1675 | 0.4982 | 0.4574 | 0.1520 | 0.9177 | 0.8131 |
| MMCL | 0.2663 | 0.0899 | 0.1071 | 0.0624 | 0.1889 | 0.8867 | 0.8310 |
| TIP | 0.3613 | 0.4956 | 0.1250 | 0.1801 | 0.1611 | 0.9800 | 0.8021 |
| DAFT | 0.3093 | 0.4175 | 0.1195 | 0.1943 | 0.1718 | 0.9711 | 0.8275 |
| MAX | 0.3652 | 0.6520 | 0.1205 | 0.1474 | 0.2021 | 0.9437 | 0.8291 |
| Concat | 0.3688 | 0.6170 | 0.1167 | 0.0340 | 0.2433 | 0.9412 | 0.8307 |
| Mul | 0.3784 | 0.5665 | 0.0459 | 0.0234 | 0.1999 | 0.9599 | 0.8271 |
| Table-LLaVA-3B | 0.0194 | 0.1682 | 0.1058 | 0.5217 | 0.2546 | 0.3500 | 0.8259 |
| Qwen3-VL-8B-Instruct | 0.2068 | 0.5775 | 0.1548 | 0.5644 | 0.2616 | 0.9615 | 0.8221 |

*Table 8.* Model performance on Regression Task

*(a)* MAE Results on Regression Datasets

| **MAE** | LOS | Respiratory Rate | Pawpularity | Anime |
|---|---|---|---|---|
| ResNet-50 | 11.107 | 3.364 | 14.934 | 0.660 |
| ViT-16 | 10.858 | 3.349 | 15.800 | 0.694 |
| LightGBM | 10.027 | 4.309 | 15.776 | 0.368 |
| TabPFN v2 | 18.111 | 3.042 | 15.383 | 0.357 |
| FT-transformer | 10.918 | 3.393 | 15.739 | 0.397 |
| CHARMS | 11.551 | 3.367 | 14.570 | 0.733 |
| MMCL | 11.483 | 3.437 | 15.192 | 0.679 |
| TIP | 13.979 | 4.208 | 15.672 | 0.410 |
| DAFT | 10.876 | 3.213 | 14.991 | 0.455 |
| MAX | 10.454 | 3.264 | 14.976 | 0.400 |
| Concat | 10.701 | 3.281 | 15.179 | 0.405 |
| Mul | 11.450 | 267970.4 | 20.739 | 0.598 |
| Qwen3-VL-8B-Instruct | 11.077 | 3.278 | 14.710 | 0.322 |
| Table-LLaVA-3B | 10.840 | 4.637 | 15.573 | 0.331 |

*(b)* $R^2$ Results on Regression Datasets

| $R^2$ | LOS | Respiratory Rate | Pawpularity | Anime |
|---|---|---|---|---|
| ResNet-50 | 0.011 | -0.003 | -0.037 | 0.161 |
| ViT-16 | 0.026 | 0.001 | 0.005 | 0.157 |
| LightGBM | 0.102 | -7.408 | 0.000 | 0.724 |
| TabPFN v2 | 0.039 | 0.124 | -0.001 | 0.724 |
| FT-transformer | 0.044 | 0.003 | -0.001 | 0.669 |
| CHARMS | 0.057 | -0.005 | 0.083 | 0.011 |
| MMCL | -0.132 | -0.009 | -0.005 | 0.109 |
| TIP | -0.355 | -0.130 | -0.041 | 0.653 |
| DAFT | 0.003 | 0.010 | -0.051 | 0.574 |
| MAX | 0.072 | -0.014 | -0.046 | 0.663 |
| Concat | 0.088 | -0.029 | -0.011 | 0.659 |
| Mul | 0.073 | -1.43e14 | -150.882 | -73.387 |
| Qwen3-VL-8B-Instruct | -0.087 | -0.012 | -0.081 | 0.774 |
| Table-LLaVA-3B | -0.043 | -1.324 | -0.228 | 0.745 |

*Table 6.* Hyperparameter Grid

*(a)* Hyperparameter Grids of Tabular Model

| Model | Hyperparameter | Values |
|---|---|---|
| **LightGBM** | Num Leaves | {31, 127} |
| | Learning Rate | Uniform{0.01, 0.1} |
| | Min Child Samples | {20, 50, 100} |
| | Min Sum Hessian In Leaf | Loguniform{1e-3, 1e-2, 1e-1} |
| | Feature Fraction | Uniform{0.8, 0.9} |
| | Bagging Fraction | Uniform{0.8, 0.9} |
| **TabTransformer** | N Blocks | {1, 2, 3, 4} |
| | Ffn D Hidden | {64, 128, 256, 512} |
| | Residual Dropout | Uniform{0.0, 0.2} |
| | Attention Dropout | Uniform{0.0, 0.5} |
| | Ffn Dropout | Uniform{0.0, 0.5} |

*(b)* Hyperparameter Grids of Vision Model

| Model | Hyperparameter | Values |
|---|---|---|
| **ResNet-50** | Lr Eval | Loguniform$\{3 \times 10^{-3}, 3 \times 10^{-4}, 3 \times 10^{-5}\}$ |
| | Weight Decay | Loguniform$\{1 \times 10^{-4}, 1.5 \times 10^{-6}\}$ |
| | Batch Size | {32, 64, 128} |
| **ViT-B/16** | Learning Rate | Loguniform{5e-4, 1e-3, 3e-3} |
| | Weight Decay | Loguniform{0.03, 0.1, 0.3} |
| | Epochs List | {50, 100, 500} |

*(c)* Hyperparameter Grids of Multi-modal Model

| Model | Hyperparameter | Values |
|---|---|---|
| **CHARMS** | Batch Size | {32, 64, 128} |
| | Lr Eval | Loguniform{1e-3, 1e-4, 1e-5} |
| **MMCL** | Lr Eval | Loguniform$\{3 \times 10^{-3}, 3 \times 10^{-4}, 3 \times 10^{-5}\}$ |
| | Weight Decay | Loguniform$\{1 \times 10^{-4}, 1.5 \times 10^{-6}\}$ |
| | Batch Size | {32, 64, 128} |
| **TIP** | Lr Eval | Loguniform$\{3 \times 10^{-3}, 3 \times 10^{-4}, 3 \times 10^{-5}\}$ |
| | Weight Decay | Loguniform$\{1 \times 10^{-4}, 1.5 \times 10^{-6}\}$ |
| | Batch Size | {32, 64, 128} |
| **MAX** | Lr Eval | Loguniform$\{3 \times 10^{-3}, 3 \times 10^{-4}, 3 \times 10^{-5}\}$ |
| | Weight Decay | Loguniform$\{1 \times 10^{-4}, 1.5 \times 10^{-6}\}$ |
| | Batch Size | {32, 64, 128} |
| **Concat** | Lr Eval | Loguniform$\{3 \times 10^{-3}, 3 \times 10^{-4}, 3 \times 10^{-5}\}$ |
| | Weight Decay | Loguniform$\{1 \times 10^{-4}, 1.5 \times 10^{-6}\}$ |
| | Batch Size | {32, 64, 128} |
| **MUL** | Lr Eval | Loguniform$\{3 \times 10^{-3}, 3 \times 10^{-4}, 3 \times 10^{-5}\}$ |
| | Weight Decay | Loguniform$\{1 \times 10^{-4}, 1.5 \times 10^{-6}\}$ |
| | Batch Size | {32, 64, 128} |

