# OpenReview forum: "VT-Bench: A Unified Benchmark for Visual-Tabular Multi-Modal Learning"
_ICML.cc/2026/Conference — ICML 2026 regular_

### Official Review · Reviewer_cB9A · 2026-03-08

**Soundness:** 2
**Presentation:** 3
**Significance:** 3
**Originality:** 2
**Overall Recommendation:** 3
**Confidence:** 4

**Summary:**

This paper introduces VT-Bench, a benchmark for visual-tabular multimodal learning, designed to address the gap in fields like healthcare where images and structured tables provide complementary information. The benchmark integrates over 756,000 samples from 14 datasets across nine domains and evaluates 21 models under two paradigms: discriminative prediction and generative reasoning. Beyond standard metrics, the authors propose two diagnostic metrics, Modal Contribution Ratio (MCR) and Modality Informativeness Ratio (MIR), to analyze fusion dynamics. The works provides some empirical insights and practical recommendations to guide future research toward more robust visual-tabular foundation models.

**Compliance With Llm Reviewing Policy:**

Affirmed.

**Final Justification:**

Soundness: The technical soundness is limited, the model evaluation methods used are not entirely appropriate, and the experimental design is not very comprehensive. Therefore, the experimental results obtained are insufficient to support their obtained conclusions.

Presentation: The overall presentation is easy to understand and has a reasonable structure. It also explains the differences between this study and previous literature.

Significance: This paper explore a significant issue: the understanding of complementary information between tables and images using multimodal models. This has considerable practical value and promising applications in real-world scenarios such as healthcare. Other researchers and practitioners may also need high-quality image-table benchmarks to measure the performance of large multimodal models, and even to obtain more robust large multimodal models.

Originality: The image-table task is not a completely new research topic. This study integrated and modified existing datasets to obtain a benchmark. This benchmark was used to evaluate existing models and methods, providing insights and suggestions for training models. However, the benchmark's representation in a "table" format is limited in scope and not entirely comprehensive.

The rebuttal reinforced my prior assessment, but I still would like improve my score, based on ICML's principles of "being considerate" and "not discouraging enthusiasm of authors".

The reason is as follows:
1. The paper claims to serve as a comprehensive benchmark. Furthermore, given that the benchmark was constructed by collecting, organizing, and integrating existing datasets, there is no justification for excluding tables in image format—or indeed, tables in any other format.
2. The author counters that only EHRXQA Stage 2 contains medical data, but it seems to me that it includes more than just EHRXQA. As seen in D.1.1, it also includes data on: Breast Cancer, Skin Cancer, MIMIC-IV v2.2, and MIMIC-CXR v2.0.0. The same question arises: Is a benchmark designed to evaluate the performance of vision-tabular models, which encompasses a wide range of medical tasks and problems, capable of yielding reliable evaluation results for vision-tabular performance?
3. Negative transfer is somewhat counter-intuitive; it feels as though it negates all the work conducted within the field of multi-modal fusion. However, I am indeed not entirely certain about this point.

**Key Questions For Authors:**

1. I believe tabular data cannot be claimed as a modality. In multimodal learning, text, images, and audio are modalities. Tabular data can be represented using text modalities (CSV, HTML, SQL) or image modalities. However, the paper consistently treats tables as a single modality in the presentation, which is confusing. Furthermore, from this perspective, a unified benchmark needs clear splits of tabular representations to differentiate the performance of image-based, CSV-based, and SQL-based tables. This benchmark only includes text-based tables, lacking image-based table representations, and doesn't provide detailed splits for CSV, HTML, and SQL text. The benchmark primarily focuses on the medical domain and is not very comprehensive; it mainly integrates and improves existing datasets, representing some incremental research.

2. The study yielded a negative transfer insight: multimodal fusion performance actually declines compared to a single-modal baseline. This contradicts the general consensus that multimodal fusion outperforms single-modal fusion. Therefore, rigorous experiments are needed to demonstrate this. One method (ablation experiments) involves assessing the performance of the image and text modules in a TIP model, rather than directly comparing a strong single-modal baseline model with a weak multimodal model. Another method is to retrain the multimodal model using the same strong single-modal model as a backbone, and then compare the single-modal and multimodal performance.

3. The study examined whether the tested models had been trained or fine-tuned, noted the scarcity of medical data in the evaluation set, questioned whether the models had encountered such data before, and assessed the reliability of the evaluation results. Specifically, in Table 2, for the generative inference task, the study opted not to fine-tune the various large models, directly evaluating them using general-purpose models. Therefore, regarding medical data like EHRXQA, had the large models encountered such data? If not, it's impossible to determine the effectiveness of utilizing complementary information from the image-table; the conclusion is that these models cannot handle medical data well. Furthermore, for the discriminative task, were the models trained on the training sets of various datasets? If so, detailed training information is lacking.

4. For multi-stage tasks, can the correct perception results from stage 1 be input into stage 2 to evaluate inference performance, thereby obtaining the model's true inference performance?

5. In rows 346-353, what does "removing both modalities" mean? Does it mean both images and tables have been removed?

6. In the benchmark, should each question be designed so that a conclusion can be reached using only images, or only using only tables, or should both images and tables be used to arrive at the answer? This requires manual sampling to check the proportion of each type of question.

7. There is a lack of analysis and ranking of the performance of each model, similar to a radar chart.

**Limitations:**

The authors have discussed their limitations in the limitation section. To add, a comprehensive and unified benchmark should include richer tabular representations, such as image-based tables, and a wider range of dataset domains. The soundness of the experiments described in the article also needs to be improved. For details, please refer to "Key Questions For Authors".

**Strengths And Weaknesses:**

Soundness: The technical soundness is limited, the model evaluation methods used are not entirely appropriate, and the experimental design is not very comprehensive. Therefore, the experimental results obtained are insufficient to support their obtained conclusions.

Presentation: The overall presentation is easy to understand and has a reasonable structure. It also explains the differences between this study and previous literature.

Significance: This paper explore a significant issue: the understanding of complementary information between tables and images using multimodal models. This has considerable practical value and promising applications in real-world scenarios such as healthcare. Other researchers and practitioners may also need high-quality image-table benchmarks to measure the performance of large multimodal models, and even to obtain more robust large multimodal models.

Originality: The image-table task is not a completely new research topic. This study integrated and modified existing datasets to obtain a benchmark. This benchmark was used to evaluate existing models and methods, providing insights and suggestions for training models. However, the benchmark's representation in a "table" format is limited in scope and not entirely comprehensive.

---

> ### Author Rebuttal · Authors · 2026-03-30
>
> Dear Reviewer cB9A,
>
> Thank you for the detailed comments.
>
> **Q1: Tabular as a modality?**
>
> We respectfully disagree that tabular data cannot be treated as a modality in our setting. This concern conflates modality with representation format. CSV, HTML, or table images are only different serializations of the same underlying tabular information. We treat tables as a distinct modality because tabular data is itself a distinctive and important data form. In domains such as healthcare and industry, it is a primary information source and has long been studied as a distinct modality type, so this is not a notion introduced by our work alone.
>
> Tabular data also differs fundamentally from text or images and poses its own challenges. It is highly heterogeneous across columns, lacks the natural inductive biases of text/images, and often relies on implicit cross-column interactions. In reasoning tasks, models must understand both the 2D structure of the table and its semantic schema. Simply flattening tables into plain text and passing them to an LLM often performs poorly [1]. Thus, we believe it is important to study tabular data as a modality distinct from text and images.
>
> In VT-Bench, prediction uses CSV inputs, while reasoning uses Markdown-serialized tables. We intentionally do not render tables as images for reasoning (the reasons in Sec. 4.2.2, Lines 268–274). VT-Bench does not provide a representation-wise breakdown across CSV/HTML/SQL, because its goal is to benchmark vision + structured tabular information under a unified protocol rather than compare all table formats separately.
>
> **Q2: Benchmark scope/novelty.**
>
> We would like to clarify that VT-Bench spans 9 domains, including pets and media, so it is not limited to medicine. Beyond integrating existing datasets, we also construct four new tasks to fill important gaps, especially in clinical prediction and vision-tabular reasoning requiring table-structure understanding and constrained numerical reasoning.
>
> **Q3: Evidence for negative transfer.**
>
> Our paper makes two empirical observations for discriminative prediction. First, multimodal models do not consistently outperform the strongest unimodal baselines (Table 2). Second, we observe negative transfer via MCR, where replacing one modality with a non-informative reference can improve performance, indicating interference rather than useful evidence. Thus, the paper already includes the ablation-style analysis requested by the reviewer. More broadly, we interpret this as evidence that vision-tabular fusion is harder to optimize reliably than vision-text fusion because the modalities are more heterogeneous.
>
> **Q4: Training details.**
>
> For reasoning, we evaluate zero-shot VLMs using official checkpoints without task-specific fine-tuning. The reviewer’s concern mainly applies to EHRXQA Stage 2; accordingly, our conclusion there is only that current VLMs perform poorly on zero-shot clinical vision-tabular reasoning (Lines 398–401), rather than directly supporting conclusions about complementary-information utilization. For prediction, the models are trained on the corresponding training splits, and the detailed setup is provided in Appendix F.1.
>
> **Q5: Stage-2 evaluation.**
>
> Yes. We already include this decoupled evaluation for EHRXQA. As stated in Appendix D.2.1, Stage 2 is evaluated given correct retrieval/perception results, while Full reports the end-to-end pipeline accuracy. This design is intended to separate retrieval/planning errors from downstream reasoning errors.
>
> **Q6: “Removing both modalities” mean?**
>
> It means replacing both image and tabular embeddings at inference time with non-informative references (i.e., the corresponding training-set mean embeddings).
>
> **Q7: Question-type proportions.**
>
> For prediction, there is no explicit oracle rule specifying which modality is strictly necessary for each label, since prediction is based on statistical dependence rather than question answering. We manually inspected the image and tabular features of each dataset and found that both modalities provide meaningful evidence (e.g., in Adoption, pet appearance from the image and attributes such as health condition/adoption fee from the table are both relevant). For reasoning, we explicitly analyzed the proportion of questions answerable from image only, table only, or both. We provide this breakdown at [link](https://anonymous.4open.science/r/icmlrebuttal-73A6/data_ratio.pdf).
>
> **Q8: Model Ranking.**
> Thank you for this useful suggestion. We have added a heatmap and radar-style ranking summary for model performance on both prediction and reasoning tasks at [link](https://anonymous.4open.science/r/icmlrebuttal-73A6/model_rank.pdf). The summary shows that strong unimodal models remain highly competitive in prediction, while Gemini-3-Flash-Preview shows the clearest advantage in reasoning.
>
> [1] Jin, Neng-Zheng, et al. A survey on table question answering: recent advances. arXiv preprint arXiv:2207.05270, 2022.

---

> > ### Author Rebuttal · Reviewer_cB9A · 2026-04-04
> >
> > I thank the authors for their rebuttal. I have carefully read all the responses. However, The response did not resolve my confusion. Therefore, I have decided to maintain my original score. I feel very sorry about that.
> >
> > My detailed reasoning is as follows:
> > 1. **Comprehensive for VT-Bench:** The authors have clarified that tabular data can be treated as a distinct modality and that VT-Bench uses CSV for prediction and Markdown for reasoning, intentionally excluding table images. However, my core concern remains: a truly comprehensive vision‑tabular benchmark should systematically account for different tabular representations (CSV, HTML, SQL, table images) and provide clear splits to assess how representation format affects performance. To put it more clearly, a comprehensive benchmark should look something like this: CSV: dataset1, dataset1, ..., SQL: dataset3, dataset4, ..., HTML: dataset5, dataset6, ..., Image: dataset7, dataset8, .... The authors acknowledge that these are merely different serializations, yet they omit table‑image based inputs and even frame this omission as a strength, which undermines the claimed "comprehensive".
> >
> > 2. **Reliable Evaluation Results:** For domain-specific data such as medical datasets, if the evaluated models have never seen such data during pre‑training, then poor zero‑shot performance on VT‑Bench could stem from two distinct sources: (1) lack of clinical reasoning ability (i.e., understanding medical concepts and logic), or (2) fundamental weakness in vision‑tabular multi‑modal processing (i.e., aligning and integrating chest X‑rays with structured EHRs). The authors’ rebuttal states that their conclusion is only that current VLMs perform poorly on zero‑shot clinical vision‑tabular reasoning (Lines 398‑401). Simply restating the conclusion does not address whether the poor performance is primarily due to missing medical knowledge or due to a general difficulty in vision‑tabular alignment. Is a benchmark designed to evaluate the performance of vision-tabular models, which encompasses a wide range of such tasks and problems, capable of yielding reliable evaluation results for **vision-tabular** performance?
> >
> > 3. **The ablation comparison in Table 2 is unfair:** I did not see the experimental results I wanted to see. It is unreasonable to directly compare multimodal models with the strongest unimodal baselines. To put it clearly, Visual Model A achieves a performance of a% on VT-Bench, while Text Model B achieves b%. By employing multimodal fusion techniques, whether through early fusion, late fusion, or integration via a large-scale multimodal model, these two models yield Model C (A+B), which demonstrates a performance of c% on VT-Bench. A meaningful assessment of Model C's performance can only be made by comparing it against Models A and B, rather than against the best-performing baseline model. This criticism is like: you ask a surveyor who holds a ruler in each hand to measure a length, then insist on comparing the result with "the most accurate single ruler in history," while refusing to compare it with the results from either the left-hand ruler or the right-hand ruler alone, which is essentially forcing the conclusion that "two hands working together are worse than a single-hand champion," ignoring what each hand could achieve on its own.

---

> > > ### Author Response · Authors · 2026-04-05
> > >
> > > Dear Reviewer cB9A,
> > >
> > > Thank you for the careful follow-up. We now better understand the remaining concerns and clarify the three main points below.
> > >
> > > **Q1: Scope of tabular representation coverage.**
> > >
> > > We agree that VT-Bench does not exhaustively cover tabular representations such as CSV, HTML, SQL, and table images under separate format-wise splits. Our goal is more focused. VT-Bench is intended as a unified benchmark for **vision + structured tabular information** across both prediction and reasoning, rather than a benchmark for systematically comparing all table serialization formats.
> > >
> > > Our benchmark in fact covers multiple underlying structured formats, including CSV-style prediction data, SQL-based EHRXQA, HTML tables in MMQA, and Markdown-style tables in DVM-Car QA. However, because our emphasis is on evaluating model capability across tasks rather than the effect of serialization format itself, we standardize reasoning inputs into Markdown whenever possible, so as to reduce additional variance introduced by format differences. This choice is also practically well motivated, since Markdown serialization is a common and robust default for table reasoning, with good readability and relatively low token overhead [1, 2, 3]. Table-image inputs are intentionally excluded because, as discussed in Sec. 4.2.2, they would introduce OCR and visual-text-recognition confounds and make large tables harder to evaluate fairly.
> > >
> > > At the same time, we would like to clarify what we mean by “comprehensive” in the paper. We do not mean exhaustive coverage of all table formats. Rather, we mean broad coverage across domains, task paradigms, model families, and evaluated capabilities within vision-tabular learning.
> > >
> > > We will revise the wording in the paper to make this scope clearer. We will also state more explicitly that richer coverage of table formats, including table images and other serializations, is valuable future work.
> > >
> > > [1] Li et al. LongTableBench: benchmarking long-context table reasoning across real-world formats and domains. Findings of ACL: EMNLP, 2025.
> > >
> > > [2] Dong et al. Encoding Spreadsheets for Large Language Models. EMNLP, 2024
> > >
> > > [3] Sui et al. Table Meets LLM: Can Large Language Models Understand Structured Table Data? A Benchmark and Empirical Study, WSDM 2024.
> > >
> > > **Q2: Evaluation reliability.**
> > >
> > > We agree that, in zero-shot clinical reasoning, poor performance may reflect both limited medical knowledge and difficulty in vision-tabular alignment. However, this concern applies only to a limited part of our benchmark, rather than to the benchmark as a whole.
> > >
> > > Specifically, this risk mainly concerns **EHRXQA Stage 2**. Our other reasoning datasets, such as DVM-Car QA and MMQA, are non-clinical and are not confounded by missing domain-specific knowledge in the same way. For prediction tasks, all evaluated models are trained on the corresponding training splits, so this issue does not arise there.
> > >
> > > Accordingly, our three key findings are not based on EHRXQA alone, but on the broader set of reliable prediction and non-clinical reasoning results in VT-Bench. The EHRXQA result should be interpreted more narrowly. We view it as a dataset-specific observation showing that current VLMs perform poorly on this particular zero-shot clinical vision-tabular task, rather than as stand-alone evidence for a general conclusion about complementary-information utilization.
> > >
> > > **Q3: Fairness of the negative-transfer comparison.**
> > >
> > > Thank you for raising this point. We would like to clarify that our paper reports two different analyses with different roles.
> > >
> > > **Comparison A: overall benchmark performance across model families**. Table 2 is intended to compare the overall performance of all evaluated models under a unified protocol, including unimodal baselines, specialized vision-tabular models, and VLMs. Comparing multi-modal models against strong unimodal baselines is meant to show their relative competitiveness on each dataset, not to serve as direct evidence for negative transfer.
> > >
> > > **Comparison B: within-model MCR ablation for modality contribution analysis**. Our actual evidence for negative transfer comes from MCR, which is computed by taking the **same multi-modal model** and replacing one modality with a non-informative reference at inference time, then measuring the resulting error change. If removing one modality improves performance, this indicates that the added modality is interfering for that model. Therefore, the ablation-style analysis requested by the reviewer is exactly the analysis reported in **Table 3**, where we found that negative transfer appears in many model-dataset pairs. This is a within-model “A+B vs. ablated A / ablated B” diagnostic under the same fusion model, rather than a comparison only against the strongest unimodal baseline. We will revise the wording to make the distinction between overall benchmark comparison and within-model MCR analysis more explicit.

---

### Official Review · Reviewer_1b6w · 2026-03-12

**Soundness:** 3
**Presentation:** 3
**Significance:** 3
**Originality:** 3
**Overall Recommendation:** 4
**Confidence:** 4

**Summary:**

This paper proposes VT‑Bench, a unified benchmark designed to evaluate vision–tabular multi‑modal learning across both discriminative prediction and generative reasoning tasks.

The benchmark aggregates 14 datasets spanning 9 domains with more than 756K samples. VT‑Bench evaluates 21 representative models, including unimodal baselines, specialized vision–tabular architectures, and general‑purpose vision‑language models (VLMs).

Beyond standard metrics, the paper introduces two diagnostic measures, including Modality Contribution Ratio (MCR) and Modality Informativeness Ratio (MIR), to quantify modality reliance.

**Compliance With Llm Reviewing Policy:**

Affirmed.

**Final Justification:**

The authors have addressed my previous concerns and I recommend Weak Accept.

**Key Questions For Authors:**

see weakness

**Limitations:**

yes

**Strengths And Weaknesses:**

**Strength**
- **Strong empirical study.** The evaluation is extensive and systematic, with appropriate breakdowns across datasets and task types. Extensive experiments show that (i) multimodal fusion often degrades performance relative to unimodal baselines, (ii) VLMs suffer from severe perception and grounding failures in vision–tabular settings, and (iii) constrained numerical and symbolic reasoning remains a major bottleneck.
- **New diagnostic metrics** The introduction of MCR and MIR is interesting. The proposed metrics addresses the limitation of existing metrics by quantifying negative transfer. The metrics could potentially help explain why multimodal models fail.
- **Presentation.** The paper is well-written and easy to understand. The benchmark is well documented, and it seems straightforward to extend if possible.


**Weakness**
- **Lack of tool-use / agentic MLLMs.** The evaluation does not consider multimodal LLMs with tool use or agentic capabilities (e.g., executable function calls for table operations, SQL engines, or external calculators). Given that many of the identified failure modes, such as numerical reasoning and multi-step table, are the settings where tool-use models could help, the benchmark may underestimate the achievable performance of modern vision–tabular systems. As a result, some conclusions about the insufficiency of prompt-based reasoning may be specific to non-agentic models rather than general limitations of vision–tabular learning.
- **Generative reasoning evaluation is limited to exact match.** The paper uses exact‑match accuracy for generative tasks. However, it could underestimate partial reasoning success. In complex QA settings, models may retrieve correct results but fail on output formatting or minor phrasing differences. Additional qualitative error analysis would strengthen the conclusions.
- **Lack of analysis of diagnostic metrics.** While MCR and MIR are intuitive and empirically useful, the authors may want to include further analysis to strengthen the contribution. For example, how sensitive are these metrics to the choice of reference (mean embedding)? The stability seems also dependent on the size of the training data?

---

> ### Author Rebuttal · Authors · 2026-03-30
>
> Dear Reviewer 1b6w,
>
> Thank you for the constructive comments. We address each concern below and will incorporate the corresponding clarifications in the final version.
>
> **Q1: Tool-use / agentic MLLMs.**
>
> We agree that tool-augmented / agentic MLLMs are an important extension, especially for operator-heavy tasks such as numerical reasoning and multi-step table operations. To further probe this setting, we added supplementary evaluations of two tool-use methods, StructGPT and Thyme. The results suggest a more nuanced picture than simply assuming that tools solve the reasoning problem. Tool use can bring localized gains on some sub-tasks, but it does not remove the main bottlenecks in VT-Bench. In particular, on DVM-Car QA, the tool-use models do not show the expected advantage on the hardest numerical sub-tasks, suggesting that the dominant failures often arise before execution, namely in cross-modal grounding, row/cell retrieval, and filtered-state construction. Similar behavior appears on MMQA, where tool use helps some sub-tasks but not overall average performance, and on EHRXQA, where the performance gap remains substantial. We therefore agree that our original wording should be scoped more carefully. The paper primarily diagnoses the limitations of prompt-only non-agentic reasoning, while these pilot results suggest that several key bottlenecks persist even with off-the-shelf tool-augmented methods. Detailed experimental results are provided in our response to Reviewer sumA (Q1).
>
> **Q2: Exact-match limitation.**
>
> We use exact-match accuracy because our generative tasks are designed with relatively canonical short answers (e.g., row index, attribute value, count, mean, or SQL-derived result), making exact match a clean and reproducible metric under a unified protocol. To reduce underestimation caused by superficial formatting differences, we apply an answer-cleaning procedure (e.g., case normalization and removal of extra spaces/invisible characters). In addition, we performed manual spot-checks on sampled predictions from each dataset and did not find evidence that minor formatting or phrasing differences are a major source of error. Therefore, while exact match is strict, we believe it is an appropriate metric here and does not materially affect our main conclusions. We will clarify this more explicitly in the revision.
>
> **Q3: Additional analysis of diagnostic metrics.**
>
> For reference sensitivity, we added alternative tests for both diagnostics. For MCR, using alternative non-informative references (zero/min/median embeddings) leaves the main qualitative conclusion unchanged, and negative transfer still appears across all tested choices. For MIR, using alternative proxy definitions (Top-2, Avg-all, Median) preserves the overall modality-direction trends. Please see Reviewer k2hS (Q1) for MCR details and Reviewer sumA (Q2) for MIR details.
>
> For training-data size stability, we further tested mean-embedding MCR using reduced training subsets (75%, 50%, and 25%) on three representative datasets (Adoption, CelebA, LOS) and four models (TIP, Concat, MAX, DAFT).
>
> | reference | Models | Adoption(I/T) | CelebA(I/T) | LOS(I/T)     |
> | --------- | ------ | ------------- | ----------- | ------------ |
> | Mean(75%) | TIP    | 0.00/100.00   | -0.64/99.36 | 14.30/85.70  |
> |           | Con    | -15.58/84.42  | 61.70/38.30 | -13.32/86.68 |
> |           | MAX    | 0.00/100.00   | 69.06/30.94 | 18.61/81.39  |
> |           | DAFT   | -80.25/19.75  | 50.72/49.28 | -0.45/99.55  |
> | Mean(50%) | TIP    | 0.00/100.00   | -0.64/99.36 | 15.36/84.64  |
> |           | Con    | -16.88/83.12  | 61.40/38.60 | -14.36/85.64 |
> |           | MAX    | 0.00/100.00   | 69.22/30.78 | 18.59/81.41  |
> |           | DAFT   | -80.25/19.75  | 50.49/49.51 | -0.37/99.63  |
> | Mean(25%) | TIP    | 0.00/100.00   | -0.64/99.36 | 18.10/81.90  |
> |           | Con    | -14.29/85.71  | 61.75/38.25 | -12.13/87.87 |
> |           | MAX    | -2.04/97.96   | 68.88/31.12 | 18.50/81.50  |
> |           | DAFT   | -80.25/19.75  | 50.65/49.35 | -0.11/99.89  |
>
> The resulting MCR signs are highly stable, with 100.0% sign agreement relative to the full-data mean embedding and average Spearman ρ = 0.99. These results suggest that MCR is highly stable across training-data sizes in our tested settings.

---

> > ### Author Rebuttal · Reviewer_1b6w · 2026-04-03
> >
> > The rebuttal addressed my main concerns. Please include the discussions and experiments to the main text to strengthen the submission.

---

> > > ### Author Response · Authors · 2026-04-03
> > >
> > > Thank you very much for the positive acknowledgment and for your helpful suggestion. We are glad that our rebuttal addressed your main concerns. We will incorporate the added discussions and supplementary experiments into the main text to strengthen the final version.

---

### Official Review · Reviewer_k2hS · 2026-03-12

**Soundness:** 2
**Presentation:** 3
**Significance:** 2
**Originality:** 3
**Overall Recommendation:** 4
**Confidence:** 3

**Summary:**

This paper introduces VT-Bench, a unified benchmark for vision–tabular multimodal learning spanning both discriminative prediction and generative reasoning. The benchmark covers 14 datasets across 9 domains with over 756K samples, evaluates 21 models, and includes two proposed diagnostics, MCR and MIR, for analyzing modality dependence and informativeness. On the reasoning side, it includes EHRXQA, MMQA, and a newly constructed DVM-Car QA benchmark. The empirical study suggests that multimodal fusion does not consistently outperform strong unimodal baselines, and that current VLMs remain limited on long-context visual grounding, structured table understanding, constrained numerical reasoning, and SQL-style clinical reasoning.

**Compliance With Llm Reviewing Policy:**

Affirmed.

**Final Justification:**

My final recommendation is Weak Accept. The paper studies a relevant and reasonably original benchmark setting for vision–tabular multimodal learning, and its scale, task coverage, and diagnostic perspective are meaningful. My main concerns were about the validation of the proposed diagnostics, the scope of some interpretive claims, the lack of robustness details, and the anomalous behavior of MUL on regression tasks. The rebuttal addressed most of these concerns by providing additional validation, clarifying the intended empirical scope of the findings, and adding useful details on stability and model behavior. While some points could still be strengthened in the final version, the rebuttal improved my confidence in the paper’s soundness and led me to adjust my score upward from Weak Reject to Weak Accept.

**Key Questions For Authors:**

1. Can the authors provide additional validation for MCR and MIR, especially sensitivity to the choice of non-informative reference used in MCR? This would strengthen confidence in the diagnostic conclusions.

2. Which conclusions are intended as observations about the evaluated models and settings, and which are intended as broader claims about vision–tabular learning? Clarifying this would help calibrate the scope of the paper’s findings.

3. Did the authors run repeated trials or collect uncertainty estimates for either prediction or reasoning tasks? This would help assess how stable some of the reported differences are.

4. Can the authors clarify the source of the instability in MUL on regression datasets, and whether similar behavior was observed elsewhere?

**Limitations:**

Partially. The paper discusses several relevant limitations, including restricted generalization beyond the evaluated datasets/models and the lack of a theoretical analysis for MCR/MIR. However, the discussion could more directly address robustness issues such as sensitivity of the diagnostics and the absence of uncertainty estimates.

**Strengths And Weaknesses:**

Strengths

* The paper studies a relevant and underexplored setting. It unifies prediction and reasoning for vision–tabular learning, whereas prior benchmarks are narrower in scope or focus on only one paradigm.

* The benchmark is substantial in scale and coverage, with 14 datasets, 9 domains, and 756K samples, including newly constructed components such as DVM-Car QA for fine-grained visual–tabular reasoning.

* The paper goes beyond aggregate accuracy by introducing MCR and MIR to analyze modality reliance and relative informativeness, which adds diagnostic value to the benchmark.

* The empirical study is reasonably broad for a benchmark paper, covering unimodal baselines, specialized fusion models, and general-purpose VLMs, and the paper also provides a unified API for evaluation.

Weaknesses

* Some interpretive claims appear broader than the evidence directly establishes. In particular, discussions of fusion bottlenecks, representation conflict, and the insufficiency of prompt-based reasoning are mainly supported by descriptive comparisons rather than controlled validation.

* The proposed diagnostics are potentially useful, but their validation is limited. MCR depends on replacing a modality with a mean embedding, and the paper explicitly notes that it does not provide a theoretical analysis linking MCR/MIR to model performance.

* The empirical results would be easier to interpret with stronger robustness analysis. The paper does not report uncertainty estimates, repeated runs, or sensitivity analysis for the proposed diagnostics or for generative reasoning evaluation.

* Some result patterns would benefit from further clarification, such as the anomalous MUL regression behavior and the extent to which the reasoning conclusions reflect limitations of current prompted VLM setups versus the broader task itself.

---

> ### Author Rebuttal · Authors · 2026-03-30
>
> Dear Reviewer k2hS:
>
> Thank you for the thoughtful comments. We address each concern below and will incorporate the corresponding clarifications in the final version.
>
> **Q1:  Additional validation for MCR and MIR.**
>
> We agree that validating the reference choice is important. To strengthen the validation of MCR, we added three alternative non-informative references and re-ran the analysis, using zero embedding (all-zero vector), min embedding (a constant vector filled with the global minimum scalar of the training embedding distribution), and median embedding (a constant vector filled with the global median scalar). Given the limited rebuttal period, we conducted this additional validation on three representative datasets and four representative models. The results are summarized below.
>
> | reference | Models | Adoption(I/T) | CelebA(I/T) | LOS(I/T)    |
> | --------- | ------ | ------------- | ----------- | ----------- |
> | Zero      | TIP    | 2.96/97.04    | -94.10/5.90 | 91.70/8.30  |
> |           | Concat | -14.71/85.29  | 62.59/37.41 | 95.71/4.29  |
> |           | MAX    | 84.47/15.53   | 55.30/44.70 | 78.34/21.66 |
> |           | DAFT   | 57.73/42.27   | 68.84/31.16 | 95.69/4.31  |
> | Min       | TIP    | 2.96/97.04    | -94.10/5.90 | 91.70/8.30  |
> |           | Concat | -5.24/94.76   | 56.93/43.07 | 3.58/96.42  |
> |           | MAX    | 84.47/15.53   | 55.30/44.70 | 78.34/21.66 |
> |           | DAFT   | 58.80/41.20   | 68.20/31.80 | 11.66/88.34 |
> | Median    | TIP    | 6.43/93.57    | -94.10/5.90 | 71.02/28.98 |
> |           | Concat | 32.22/67.78   | 62.59/37.41 | 68.05/31.9  |
> |           | MAX    | 23.81/76.19   | 55.30/44.70 | 61.16/38.84 |
> |           | DAFT   | -14.81/85.19  | 67.11/32.89 | -0.02/99.98 |
>
> The main conclusion remains unchanged, and **negative transfer still appears under all tested reference choices.** While the exact magnitudes vary, the qualitative pattern is preserved across alternatives. Quantitatively, the average sign agreement between the original mean-embedding MCR and these alternatives is 63.4%, suggesting that the conclusions are not solely an artifact of the mean-embedding choice.
>
> For MIR, we additionally computed the metric under three alternative proxy definitions. These results suggest that MIR is reasonably robust to proxy definition and not driven by the specific Top-1 formulation. Detailed data and analysis are provided in our response to Reviewer sumA (Q2).
>
> We agree that MCR/MIR are empirical diagnostics rather than theoretically grounded quantities. Developing a stronger theoretical foundation for them is an important direction that we will explore in future work.
>
> **Q2: Scope of the paper’s findings.**
>
> Our three key findings are intended as empirically grounded summaries of core challenges in vision-tabular multimodal learning. They are not derived from isolated descriptive comparisons, but from recurring patterns repeatedly observed across multiple domains, task settings, and representative model families in VT-Bench. In particular, our evaluation covers several representative and competitive model categories commonly used in multimodal learning, including unimodal baselines, specialized vision-tabular models, and general-purpose VLMs. Because similar patterns emerge across these diverse settings, we believe it is reasonable to interpret the findings as highlighting core challenges in vision-tabular multimodal learning, such as negative transfer, grounding difficulties, and constrained reasoning limitations. We will revise the wording to make this empirical scope more explicit.
>
> **Q3: Repeated trials / uncertainty estimates.**
>
> Yes. For discriminative prediction, all reported results are averaged over three random seeds to account for training variability. For generative reasoning, we use fixed official checkpoints with deterministic zero-shot inference (fixed prompts, temperature = 0, greedy decoding), so there is no analogous seed variability and repeated runs would produce the same outputs under our protocol.
>
> **Q4: Source of the instability in MUL.**
>
> Our interpretation is that MUL is unstable because of its scale sensitivity. MUL performs channel-wise multiplicative fusion, so large activations or mismatched feature scales across modalities can be amplified by the product operation. This is especially problematic for regression, where outputs are unconstrained continuous values, and extreme fused representations can lead to very large errors (Appendix B, Line 679). We observe this behavior across multiple regression datasets, whereas such severe instability does not appear in classification tasks. By contrast, other late-fusion methods such as Concat and MAX remain much more stable overall.

---

> > ### Author Rebuttal · Reviewer_k2hS · 2026-04-02
> >
> > The rebuttal addressed most of my main concerns, especially by providing additional validation for the proposed diagnostics, clarifying the scope of the claims, and explaining the robustness and instability issues more clearly, so I have adjusted my score upward.

---

> > > ### Author Response · Authors · 2026-04-03
> > >
> > > Thank you very much for the positive acknowledgment and for adjusting your score upward. We are glad that our rebuttal addressed most of your main concerns. We will incorporate the added validations and scope clarifications into the final version.

---

### Official Review · Reviewer_sumA · 2026-03-13

**Soundness:** 3
**Presentation:** 3
**Significance:** 3
**Originality:** 2
**Overall Recommendation:** 4
**Confidence:** 3

**Summary:**

This paper introduces VT-Bench, a unified benchmark for vision-tabular learning, covering both prediction and reasoning tasks with image-plus-table inputs. It includes 14 datasets, 21 evaluated models, four new datasets, and two diagnostic metrics (MCR and MIR), and the experiments show that current multimodal methods still struggle: fusion models often do not reliably beat strong unimodal baselines, while VLMs remain limited in visual grounding, table understanding, and numerical reasoning.

**Compliance With Llm Reviewing Policy:**

Affirmed.

**Key Questions For Authors:**

1. In the generative reasoning setting, all models are evaluated zero-shot with Markdown-serialized tables under a consistent decoding protocol, and executable/tool-augmented reasoning is explicitly left out of scope. Could the authors clarify whether they ran any pilot comparisons with alternative table interfaces or tool-augmented settings, and whether the main failure modes reported here still persist under those variants?
2. MCR replaces one modality with the mean training embedding, while MIR uses the best unimodal error as a proxy for dataset-level modality informativeness. Could the authors provide additional evidence that the main conclusions drawn from MCR/MIR are robust to alternative ablation/reference choices or proxy definitions?
3. For the newly introduced datasets, especially the MIMIC-based clinical prediction tasks and DVM-Car QA, could the authors clarify the exact split granularity and leakage-prevention strategy?

**Limitations:**

yes

**Strengths And Weaknesses:**

Soundness: The paper is generally sound as a benchmark and empirical analysis paper. The task setup is clearly defined, and the benchmark is fairly broad, covering 14 datasets, 21 models, four newly introduced datasets, and both prediction and reasoning settings. The main empirical conclusions, such as the difficulty of reliably benefiting from multimodal fusion and the limitations of current VLMs on visual grounding and constrained reasoning, are largely supported by the reported results. A limitation is that some parts of the analysis framework (e.g., MCR/MIR) are mainly empirical rather than theoretically grounded.

Presentation: The paper is clearly written and well organized. The motivation, benchmark design, and separation between discriminative prediction and generative reasoning are easy to follow, and the work is positioned clearly relative to prior benchmarks.

Significance: This is one of the stronger aspects of the paper. Vision-tabular learning is an important but under-standardized multimodal setting, and a unified benchmark spanning both prediction and reasoning could be valuable infrastructure for future work. The contribution is more significant as an evaluation and problem-definition effort than as a direct modeling advance.

Originality: The originality is moderate, primarily in benchmark construction and empirical diagnosis rather than in algorithm design. The main novelty comes from unifying multiple vision-tabular task types in one benchmark, introducing four new datasets, and proposing diagnostic metrics to study modality dependence. The paper is less original from a methods perspective, since it does not introduce a fundamentally new learning algorithm.

---

> ### Author Rebuttal · Authors · 2026-03-30
>
> Dear Reviewer sumA,
>
> Thank you for the constructive comments. We address each concern below and will incorporate the corresponding clarifications in the final version.
>
> **Q1: Executable / tool-augmented reasoning.**
>
> We agree this is an important direction. To further probe this setting, we additionally evaluated two tool-augmented / interface-based reasoning methods, StructGPT[1] and Thyme[2]. The results are summarized below:
>
> | Task         | StructGPT | Thyme |
> | ------------ | --------: | ----: |
> | **DVM-Car QA**  |           |       |
> | Ident.       | 0.721 | 0.751 |
> | Row Loc.     | 0.858 | 0.436 |
> | Attr. Ret.   | 0.496 | 0.672 |
> | Count        | 0.300 | 0.224 |
> | Cond. Mean   | 0.007 | 0.139 |
> | Avg. Acc.       | 0.415 | 0.368 |
> | **MMQA**     |           |       |
> | TableQ       | 0.718 | 0.645 |
> | TextQ        | 0.688 | 0.620 |
> | ImageQ       | 0.549 | 0.461 |
> | ImageListQ   | 0.232 | 0.174 |
> | Multi-Hop    | 0.346 | 0.427 |
> | Avg. Acc.        | 0.541 | 0.509 |
> | **EHRXQA**   |           |       |
> | Full         | 0.142 | 0.044 |
> | STAGE1           | 0.463 | 0.241 |
> | STAGE2           | 0.308 | 0.182 |
>
> Detailed data are presented in this [**link**](https://anonymous.4open.science/r/icmlrebuttal-73A6/reason.pdf).
>
> These pilot results suggest that tool-augmented / interface-based methods bring localized gains but do not remove the main bottlenecks in VT-Bench. On DVM-Car QA, these methods do not show the expected advantage on the hardest numerical sub-tasks, suggesting that the dominant failure is often not arithmetic itself but earlier grounding and retrieval errors. If these steps fail, the tool may still execute over the wrong operands. Similar behavior appears on MMQA, where these methods help some sub-tasks but not average performance, and on EHRXQA, where the gap remains substantial. We therefore revise our scope claim accordingly. The original paper diagnoses the limitations of prompt-only non-agentic reasoning, while these pilot results suggest that key bottlenecks still persist even with off-the-shelf tool-augmented / interface-based methods, especially in cross-modal grounding, table-structure grounding, and clinically realistic programmatic reasoning.
>
> **Q2: Robustness of MCR/MIR to alternative reference / proxy choices.**
>
> For MCR, we additionally tested three alternative non-informative references (zero/min/median). The main qualitative conclusion remains unchanged, and **negative transfer still appears across all tested choices**. The average sign agreement with the original mean-embedding MCR is 63.4%, suggesting that the qualitative pattern is reasonably consistent and not solely an artifact of the mean-embedding choice. Please see Reviewer k2hS (Q1) for more details.
>
> For MIR, we further considered three alternative proxy definitions: Top-2 (average of the two best unimodal errors), Avg-all (average over all unimodal errors), and Median (median unimodal error). The resulting MIR values are:
>
> |        | Adoption | Breast Cancer | Skin Cancer | DVM-Car | CelebA | Pneumonia | Infarction | Pawpularity | Anime | LOS  | RR   |
> | ------ | -------- | ------------- | ----------- | ------- | ------ | --------- | ---------- | ----------- | ----- | ---- | ---- |
> | Top-2  | 1.22     | 1.32          | 1.77        | 1.11    | 0.99   | 1.00      | 1.01       | 1.02        | 1.75  | 1.04 | 0.78 |
> | Avg    | 1.20     | 1.29          | 1.70        | 1.09    | 0.99   | 0.99      | 0.96       | 1.01        | 1.70  | 0.99 | 0.69 |
> | Median | 1.22     | 1.27          | 1.76        | 1.09    | 0.99   | 1.00      | 0.89       | 1.02        | 1.72  | 0.92 | 0.73 |
>
> Across these alternatives, the average direction agreement with the original Top-1 MIR (i.e., MIR > 1 vs. < 1) is 78.8%, and the average Spearman ρ is 0.694. These results suggest that MIR is reasonably robust to proxy definition and not driven by the specific Top-1 formulation.
>
> We agree that MCR/MIR are empirical diagnostics rather than theoretically grounded quantities. Developing a stronger theoretical foundation for them is an important direction that we will explore in future work.
>
> **Q3: Dataset split granularity and leakage prevention.**
>
> For the MIMIC-based clinical prediction tasks, we use an 8:1:1 patient-level split, so that all records from the same patient appear in only one subset, preventing subject-level leakage (Appendix D.1.2, Line 902). For DVM-Car QA, this issue is different because it is a zero-shot generative reasoning benchmark rather than a supervised train/test task; there is no model training on this dataset, so split-based leakage prevention is not applicable in the same way.
>
> [1] Jiang, Jin-Hao, et al. Structgpt: A general framework for large language model to reason over structured data. Proceedings of the 2023 Conference on Empirical Methods in Natural Language Processing, 2023.
>
> [2] Zhang, Yi-Fan, et al. Thyme: Think beyond images. arXiv preprint arXiv:2508.11630, 2025.

---

> > ### Author Rebuttal · Reviewer_sumA · 2026-04-03
> >
> > The rebuttal addressed my main questions, especially the additional checks on MCR/MIR, the pilot results on tool-augmented reasoning, and the clarification of split and leakage details. I appreciate these additions and will maintain my current score.

---

> > > ### Author Response · Authors · 2026-04-04
> > >
> > > Thank you very much for the positive acknowledgment. We are glad that our rebuttal addressed your main questions. We will incorporate the added analyses into the final version of the paper.

---

### Decision · Program_Chairs · 2026-04-30

**Decision:**

Accept (regular)

**Comment:**

This paper introduces VT-Bench, a unified benchmark for evaluating vision-tabular and vision language models spanning 14 datasets across 9 domains with over 756K samples, evaluating 21 models on both discriminative prediction and generative reasoning tasks, along with two diagnostic metrics (MCR and MIR) for analyzing modality dependence. Reviewers praised the practical significance of standardizing this underexplored evaluation setting and the diagnostic value of the proposed metrics, but raised concerns about the robustness of MCR/MIR, the absence of tool-augmented reasoning evaluation, and the inaccurate claim of comprehensiveness. During the rebuttal, the authors added tool-augmented evaluations (StructGPT, Thyme), alternative reference/proxy robustness checks for MCR/MIR, and training-data size sensitivity analysis, which fully satisfied three reviewers. Reviewer cB9A remaining concerns about tabular format exhaustiveness and medical zero-shot confounds were partially addressed by authors and assessed by Reviewer k2hS as valid but insufficient to overturn acceptance. The AC recommends acceptance contingent on revising the comprehensiveness claims, adding results and discussion from the rebuttal, and explicitly discussing the medical zero-shot limitation.